# UniGS: Unified Language-Image-3D Pretraining with Gaussian Splatting

**Haoyuan Li**[1*], **Yanpeng Zhou**[2], **Tao Tang**[1], **Jifei Song**[2], **Yihan Zeng**[2],

**Michael Kampffmeyer**[3], **Hang Xu**[2], **Xiaodan Liang**[1,4,5†]

[1]Shenzhen campus of Sun Yat-sen University, [2]Huawei Noah's Ark Lab,

[3]UiT The Arctic University of Norway, [4]Peng Cheng Laboratory,

[5]Guangdong Key Laboratory of Big Data Analysis and Processing

`https://github.com/Li-Hao-yuan/UniGS.`

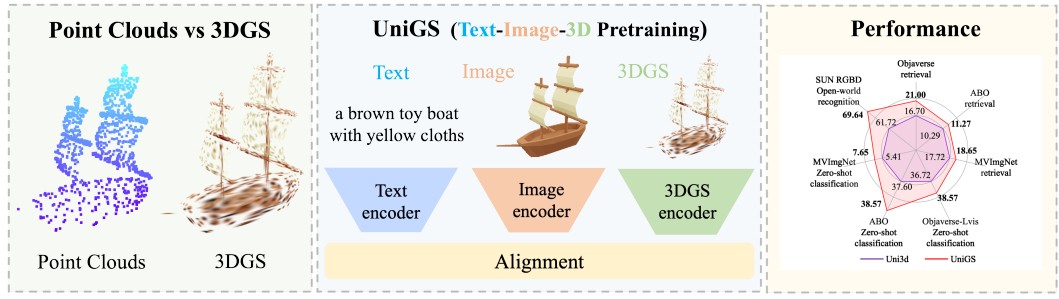

Figure 1: **Left:** information gap in different 3D representations. **Middle**: our UniGS, a novel unified text-image-3D pre-training framework, leverages 3DGS as the 3D representation. **Right:** our UniGS learns a more general and stronger multi-modal representation.

## ABSTRACT

Recent advancements in multi-modal 3D pre-training methods have shown promising efficacy in learning joint representations of text, images, and point clouds. However, adopting point clouds as 3D representation fails to fully capture the intricacies of the 3D world and exhibits a noticeable gap between the discrete points and the dense 2D pixels of images. To tackle this issue, we propose UniGS, integrating 3D Gaussian Splatting (3DGS) into multi-modal pre-training to enhance the 3D representation. We first rely on the 3DGS representation to model the 3D world as a collection of 3D Gaussians with color and opacity, incorporating all the information of the 3D scene while establishing a strong connection with 2D images. Then, to achieve Language-Image-3D pertaining, UniGS starts with a pre-trained vision-language model to establish a shared visual and textual space through extensive real-world image-text pairs. Subsequently, UniGS employs a 3D encoder to align the optimized 3DGS with the Language-Image representations to learn unified multi-modal representations. To facilitate the extraction of global explicit 3D features by the 3D encoder and achieve better cross-modal alignment, we additionally introduce a novel Gaussian-Aware Guidance module that guides the learning of fine-grained representations of the 3D domain. Through extensive experiments across the Objaverse, ABO, MVImgNet and SUN RGBD datasets with zero-shot classification, text-driven retrieval and open-world understanding tasks, we demonstrate the effectiveness of UniGS in learning a more general and stronger aligned multi-modal representation. Specifically, UniGS achieves leading results across different 3D tasks with remarkable improvements over previous SOTA, Uni3D, including on zero-shot classification (+9.36%), text-driven retrieval (+4.3%) and open-world understanding (+7.92%).

---

[*]Work done as an intern at Huawei Noah's Ark Lab.
[†]Corresponding author.

# 1    INTRODUCTION

The remarkable success of 2D Image-Text pre-training through modality alignment via contrastive learning (Radford et al., 2021; Sun et al., 2023; Fang et al., 2023; Schuhmann et al., 2022; Qi et al., 2020; Changpinyo et al., 2021; Hong et al., 2021) has recently inspired a line of work pursuing 3D pre-training (Xue et al., 2023a;b; Zeng et al., 2023; Zhou et al., 2024; Liu et al., 2024; Zhang et al., 2022; Huang et al., 2023; Afham et al., 2022). Leveraging diverse large-scale 3D datasets, recent works such as (Liu et al., 2024; Zhou et al., 2024) expand the traditional 2D task to text-image-3D pertaining by including point clouds as 3D representations, resulting in considerable improvements in 3D zero-shot/open-world object detection, classification and retrieval tasks. While point clouds serve as a natural step towards 3D representations, there are inherent limitations when using them to represent 3D objects. In particular, as illustrated in Fig. 1, point clouds consist of a discrete set of points and thus struggle to accurately capture fine-grained geometric details and surface textures of common objects, limiting the performance of 3D representation learning approaches. Moreover, there exists a noticeable gap between the discrete points and the dense 2D pixels of images, which further hinders the learning of joint multi-modal representations.

On the other hand, 3D Gaussian Splatting (3DGS) (Kerbl et al., 2023) has recently revolutionized 3D scene representations, and offers a promising and more efficient alternative to facilitate 3D representation learning. Specifically, 3DGS models scenes as a set of 3D Gaussians, which effectively reconstruct the 3D target object as well as provide efficient correspondence between 3D and 2D images through the splatting rendering algorithm. Furthermore, 3DGS offers the advantage of utilizing a more diverse range of data sources as it can leverage multi-view images or COLMAP data (Yu et al., 2023; Schonberger & Frahm, 2016) for optimization with minimal overhead and collection time. Additionally, existing point cloud datasets can be used as the initialization for 3D Gaussian locations, further enhancing the capabilities of 3DGS. However, simply retraining existing multi-modal frameworks like CLIP[2] (Zeng et al., 2023) or Uni3D (Zhou et al., 2024) to leverage 3DGS as the 3D representation is not effective. As the spatial connections of 3DGS may be insufficient to capture and express objects due to the fact that 3DGS are not necessarily distributed on the surface of objects.

To this end, we propose UniGS, which leverages 3DGS as the 3D representation for unified language-image-3D pre-training and enhances the performance of 3D understanding. To better model and understand the explicit features of 3DGS, UniGS additionally proposes a novel Gaussian-Aware Guidance module. Specifically, UniGS utilizes a parallel-structure ViT as the 3D encoder consisting of a fundamental encoder and advanced encoder, where the fundamental encoder encodes spatial information together with color and the advanced encoder the spatial information with the remaining 3D Gaussian attributes, where pre-trained models can be leveraged for initialization of the fundamental encoder. With priors extracted from the fundamental encoder, the advanced encoder aggregates priors through cross-attention layers for guiding the 3DGS feature learning, unlocking the superior performance that comes with leveraging 3DGS. Through extensive experiments across the Objaverse (Deitke et al., 2023), ABO (Collins et al., 2022), MVImgNet (Yu et al., 2023) and SUN RGBD (Song et al., 2015) datasets and various tasks, we demonstrate the effectiveness of UniGS in learning a more general and stronger multi-modal representation. Specifically, UniGS achieves state-of-the-art results across different 3D tasks with remarkable improvements, including zero-shot classification (+9.36%), text-driven retrieval (+4.3%), and open-world understanding (+7.92%). Our contributions can be summarized as follows:

- We propose UniGS, a novel unified text-image-3D pre-training framework, which leverages 3DGS as the 3D representation for learning a more general and stronger multi-modal representation.

- We propose a novel Gaussian-Aware Guidance module to leverage priors from pre-trained point clouds encoders to guide the learning of the Gaussian features for better 3D understanding.

- Our proposed approach achieves state-of-the-art performance on various challenging datasets, demonstrating the effectiveness in learning strong cross-model representations.

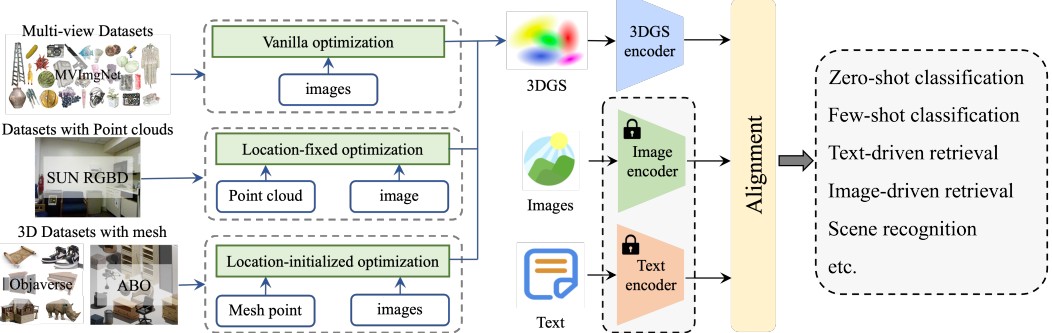

Figure 2: **The overview of UniGS.** UniGS is an innovative, unified, and scalable 3D pretraining framework designed for 3D representation learning. It offers versatile pipelines for various datasets, enabling the efficient 3DGS acquisition to enhance 3D representation learning with SoTA CLIP models. UniGS demonstrates exceptional performance across a broad range of benchmarks.

## 2 RELATED WORK

**Multi-modal pretraining via contrastive learning**. Leveraging multi-modal data to pre-train modality-specific encoders via contrastive alignment has received considerable attention in recent years (Radford et al., 2021; Mu et al., 2022) due to its ability to leverage massive in-the-wild datasets of paired data. While early works have largely been focused on image-text data, promoting research on text-based image manipulation (Patashnik et al., 2021), open vocabulary object detection (Gu et al., 2021; Gao et al., 2022), language grounding (Li et al., 2022) and zero-shot segmentation (Xu et al., 2023), there has been an increasing focus on learning 3D representations lately (Xue et al., 2023a;b; Zeng et al., 2023; Zhou et al., 2024). These approaches largely follow the contrastive learning paradigm by adding a new 3D representation encoder and aligning it with the 2D and/or text modalities. However, due to the difficulty of collecting and constructing 3D representation data, current approaches leverage point clouds. More specially, PointCLIP (Zhang et al., 2022) and CLIP2Point (Huang et al., 2023) extract depth maps from point clouds to obtain image-like data that can be leveraged for contrastive pretraining. Since these depth maps lose plenty of spatial information of the original point cloud data structure, CLIP$^2$ (Zeng et al., 2023) instead learns a 3D encoder directly on point clouds, demonstrating robustness in real-world indoor and outdoor scenarios. However, the flexibility and simplicity of point clouds come at a cost as specific shape and texture information of the object surface is lost, leading to ambiguities. In this work, we, therefore, adopt 3DGS (Kerbl et al., 2023) to replace point clouds as the 3D representation to alleviate this loss of information.

**Zero-shot/Open-world Learning in 3D.** Considerable progress has been made to project a point cloud into 3D voxels (Shi et al., 2020; Maturana & Scherer, 2015; Qi et al., 2017a) and extract features that can be associated with semantic category information. PointNet++ (Qi et al., 2017b) proposes a hierarchical neural network to extract local features with increasing contextual scales and PointMLP (Ma et al., 2022) proposes a pure residual MLP network while achieving competitive results. More recently, self-supervised learning (Yu et al., 2022; Pang et al., 2022) and unsupervised learning (Afham et al., 2022; Liang et al., 2021; Liu et al., 2022) approaches for 3D understanding have also shown promising performance. However, while the current supervision-based methods are restricted by the annotated training datasets and show poor performance on unseen categories, self-supervised and unsupervised-based methods can not directly be transferred to zero-shot tasks with open-world vocabularies due to the limited downstream annotations. Therefore, we construct a language-image-3D dataset for pretraining to learn transferable 3D representations that are aligned to an open-vocabulary language space to facilitate zero-shot transfer.

## 3 METHODOLOGY

In this section, we introduce our proposed UniGS in detail. We first review background information of 3DGS and next provide an overview of UniGS in Section 3.1. The proposed cross-modal contrastive learning framework for multi-modal alignment is then presented in Section 3.2 before we introduce the details of the Gaussian-Aware Guidance in Section 3.3. We further present details on scaling up

and initializing UniGS in Section 3.4. Finally, we present how we ensemble 3DGS datasets from existing datasets in Section 3.5.

## 3.1 PRELIMINARIES AND OVERVIEW

**3D Gaussian splatting.** 3D Gaussian splatting (Kerbl et al., 2023) proposed an explicit model design and efficient differentiable rendering implementation, which enable faster training and high-quality performance. 3DGS employs a collection of anisotropic Gaussian primitives, denoted as $G = \{g_1, g_2, \ldots, g_N\}$, to represent the scene. Each 3D Gaussian sphere $g$ can be parameterized by the following attributes: (1) located at 3D position $\mu \in \mathcal{R}^3$ (2) with color defined by SH ($3 + c \in \mathcal{R}^k$), (3) opacity $\alpha \in [0, 1]$ and a 3D shape decomposed into (4) a scaling factor $s \in \mathcal{R}^3_+$ and (5) a rotation quaternion $r \in \mathcal{R}^4$, where $k$ denotes the freedom of SH basis. Clearly, a Gaussian sphere can be characterized by a high-dimensional feature representation concatenating its position $\mu$, color $c$, opacity $\alpha$, scaling factor $s$, and rotation $R$, where

$$g = (\mu, c, \alpha, s, R). \tag{1}$$

In the rendering process, a splatting pipeline is utilized, projecting 3D Gaussians onto the 2D image plane. This projection splats the 3D Gaussians into 2D counterparts on the image plane, which are then blended using the $\alpha$-blending algorithm to determine the final color composition.

$$\mathbf{C} = \sum_{i \in N} \mathbf{c}_i \alpha_i \prod_{j=1}^{i=1} (1 - \alpha_i), \mathbf{\Sigma} = \mathbf{R}\mathbf{S}\mathbf{S}^T \mathbf{R}^T \tag{2}$$

where $c_i$ denotes the color defined by spherical harmonics (SH) (Kerbl et al., 2023) coefficients of each 3D Gaussian, $\mathbf{R}$ and $\mathbf{S}$ are the matrix representation of $R$ and $s$, $\alpha_i$ is calculated by the multiplication of a 2D Gaussian with covariance $\mathbf{\Sigma}$ and a learned per-point opacity (Yu et al., 2021).

**Overview of UniGS.** UniGS facilitates 3D multi-modal representation learning by leveraging the informative and effective 3DGS representation and supports open-world learning. The overview of UniGS is depicted in Fig. 2. With Image Encoder and Text Encoder from pre-trained Text-image aligned model (Radford et al., 2021) for a shared latent space, UniGS formulates a parallel-structure dual-branch encoder, which adopts a frozen pre-trained point cloud encoder as the fundamental encoder for priors and leverages another encoder as the advanced encoder modeling high-level information.

## 3.2 CROSS-MODAL REPRESENTATION ALIGNMENT

To align multi-modal representations of text, image, and 3D domains, UniGS adopts the pre-trained language-image model CLIP (Radford et al., 2021) to provide a common language-image latent space that serves as the target latent space to align 3DGS representation to (see Fig. 2). To facilitate the transferability of the learned representations and enable zero-shot/open-word recognition, the text and image encoders of the CLIP model which defines the common latent space are frozen. In particular, we take inspiration from the contrastive loss in (Radford et al., 2021; Zeng et al., 2023) and propose **Language-3DGS** and **Image-3DGS Alignment** losses to bridge the domain gap among the different modalities.

**Language-3DGS Alignment.** Given a text-image-3DGS triplet, $\{X_T, X_I, X_G\}$, text features, $f^T \in \mathbb{R}^{C^T}$, image features, $f^I \in \mathbb{R}^{C^I}$, and 3DGS features, $f^G \in \mathbb{R}^{C^G}$ can be obtained through the corresponding modality encoders. The contrastive loss between the text and 3D modality is then utilized to align the text and 3DGS feature representations. Let $N$ denotes the batch size and $\tau$ the temperature coefficient, the Language-3DGS Alignment training objective $L(T, G)$ can be described as:

$$L(T, G) = \frac{1}{N} \sum_{i \in N} \mathcal{L}(i, T, G) = -\frac{1}{N} \sum_{i \in N} \log \frac{\exp(f_i^T \cdot f_i^G / \tau)}{\exp(f_i^T \cdot f_i^G / \tau) + \sum_{j \in N, X_i^T \neq X_j^T} \exp(f_i^T \cdot f_j^G / \tau)} \tag{3}$$

Figure 3: **Model overview of UniGS.** Let $\mu, c, \alpha, s, R$ denote the location, color, opacity, scale, and rotation attribute of 3DGS. (a) Given a 3DGS input, the pre-trained and frozen branch takes 3DGS locations and color as input while the second branch, which is initialized from scratch, focuses on the 3DGS location and the remaining attributes. (b) shows the details of our 3D Encoder and how the prior is leveraged through cross-attention layers.

**Image-3DGS Alignment.** Similarly, we apply the contrastive loss to align the image and 3DGS features. The Image-3DGS Alignment objective $L(I, G)$ is defined as:

$$L(I, G) = \frac{1}{N} \sum_{i \in N} \mathcal{L}(i, I, G) = -\frac{1}{N} \sum_{i \in N} \log \frac{\exp(f_i^I \cdot f_i^G / \tau)}{\exp(f_i^I \cdot f_i^G / \tau) + \sum\limits_{j \in N, j \neq i} \exp(f_i^I \cdot f_j^G / \tau)}, \quad (4)$$

Following (Zeng et al., 2023), the final cross-modal contrastive learning objective $L_{CM}(T, I, G)$ can be obtained by combining the text-3DGS and image-3DGS alignment objective, namely, $L(T, G)$ and $L(I, G)$:

$$L_{CM}(T, I, G) = \lambda_1 L(T, G) + \lambda_2 L(I, G), \quad (5)$$

where both hyper-parameters, $\lambda_1$ and $\lambda_2$, are set to 0.5.

### 3.3 GAUSSIAN-AWARE GUIDANCE

Admittedly, projecting point clouds into 3D voxels with a 3D backbone (Qi et al., 2017a;b) can be helpful for understanding the relationships between global position and feature derived from non-positional information. However, we observe that the explicit feature of 3DGS will be ignored as the voxelization procedure loses the shape and texture information. To address this issue, a Transformer-based model pattern is adopted for feature-context learning and model scalability.

For better modeling and understanding of the 3D domain represented by 3DGS, we further propose a Gaussian-aware Guidance module. Specifically, this module, as highlighted in the dashed box in Fig. 3a, formulates a dual-branch 3D encoder, where the fundamental encoder $E_{fun}$ leverages the 3D ViT Encoder pre-trained on 3D point clouds from (Zhou et al., 2024) to model the low-level features $f_{fun}$ including spatial and color information of 3DGS, while the advanced encoder $E_{adv}$ additionally models the high-level features $f_{adv}$, the relationship between spatial connections and 3DGS feature. As the number of Gaussian spheres allocated to an object or scene represented by 3DGS varies, a group divider is leveraged to process the input 3DGS into a fixed number of Gaussian spheres. The CNN layers map low-dimensional raw features to a high-dimensional feature space, consistent with Uni3D. Moreover, cross-attention layers are built to extract guidance of embeddings from the fundamental encoder and improve the alignment of the 3DGS features with other domains. We denote the process as $CA$, where

$$CA = softmax(\frac{Q_{fun}K_{adv}^T}{\sqrt{d_k}})V_{adv}. \quad (6)$$

Finally, we concatenate the features of $f'_{fun}$ and $f'_{adv}$, respectively generated by the fundamental encoder and the advanced encoder, and then map it through an MLP to the same dimension as the pre-trained image-text model. Let SA be the acronym of self-attention and $f_\theta(\cdot)$ denote the process

of group divider and CNN layers. The process of encoding 3DGS features based on Gaussian-Aware Guidance can then be represented by:

$$f_{fun} = (\mu, c), f_{adv} = (\mu, \alpha, s, R) \tag{7}$$

$$f'_{fun} = SA(f_\theta(f_{fun})), f'_{adv} = CA(f_\theta(f_{adv})) \tag{8}$$

$$f^G = MLP(concat(f'_{fun}, f'_{adv})) \tag{9}$$

### 3.4 SCALING UP 3DGS BACKBONES

**Scaling Up UniGS.** Previous works have achieved good performance by designing specific model architectures and shown case effectiveness in various applications. However, these methods are either limited to a certain small-scale dataset or data sources like point clouds which are expensive to collect. Instead, with recent advances in large-scare multi-view datasets (Yu et al., 2023), our method adopts 3DGS representations and designs a ViT-based Encoder to encode the 3D modality. Therefore, our model can naturally solve the difficulties by simply scaling up the data and model size with well-studied unified scaling-up strategies.

**Initializing UniGS.** Restricted by the scale of the 3D dataset, directly pre-training each 3D backbone for specific 3D tasks leads to expensive training costs and may suffer from difficulties in convergence or overfitting. To overcome this issue, we follow Uni3D (Zhou et al., 2024) and adopt off-the-shelf pre-trained large models (Sun et al., 2023; Fang et al., 2023; Radford et al., 2021; Caron et al., 2021) with ViT-based structure in the other modalities as the initialization of the 3D backbone to transfer their rich underlying representational abilities to language-image-3D pre-training Different from Uni3D (Zhou et al., 2024), UniGS further establishes stable cross-modal contrastive learning through the Gaussian-Aware Guidance module, which introduces a new perspective of leveraging pre-trained priors for stabilizing the learning of large-scale 3D representations, and provide guidance on spatial information for understanding and aligning 3DGS representations with other modalities.

### 3.5 ENSEMBLING 3DGS DATASETS

To create 3DGS-text-image triplets for training, we over-sample points from the mesh surface uniformly to capture the details of each object. We random sample $N$ point clouds to initialize the 3D Gaussians, where $N$ is set to 1024 in Objaverse, ABO, and MVImgNet. Next, we use rendered images for 3DGS optimization. More specifically, for ABO (Collins et al., 2022), we uniformly render 72 images covering the whole shape, while we obtain the rendered image from (Liu et al., 2023) for the Objaverse dataset. Finally, we collect and clean the caption for each dataset. We clean the given caption of ABO and sort ABO objects into 23 classes with 7929 items, while human-verified and machine-generated high-quality captions from (Luo et al., 2024; Dong et al., 2024) are utilized for Objaverse. During the optimization of 3DGS, we control the number of 3D Gaussians by adjusting the 3DGS optimization scheme. In particular, after each 3DGS duplicating and pruning step, we sort the 3D Gaussians by opacity and only keep the top $N$. To facilitate better optimization results for the SUN RGB-D dataset, we do not restrict $N$ to 1024 as it is under a sparser setting with a single image provided for each scene.

## 4 EXPERIMENT

In this section, we evaluate the proposed UniGS on a range of 3D understanding tasks, highlighting its ability to learn more expressive and informative 3D representations than prior approaches. In particular, we consider the tasks of object retrieval, zero-shot classification, and scene understanding. We further conduct detailed and comprehensive ablation studies to reveal the impact and power of our design for cross-modal learning.

### 4.1 EXPERIMENTAL SETUP

**Baseline.** We compare UniGS with (Zeng et al., 2023; Zhou et al., 2024; Zhang et al., 2024; Qi et al., 2023; 2024). To further understand how the 3DGS features, the pre-trained weights, and the parallel structure impact 3D representation learning, we retrain the most relevant model (Uni3D) with different settings for a fair comparison. As for the baseline model, we further report the performance of the original Uni3D model trained with point clouds and the altered version trained with the 3D location attributes (mean value of Gaussians) of 3DGS instead of point clouds for completeness. Note that 3DGS does not necessarily exist on the surface of objects, so there is a certain difference between point clouds and the 3D location attributes of 3DGS.

| Methods | Top1 | Avg. Top5 | Top 10 | Representation | Text-image Model | Embedding dim |
|---------|------|-----------|--------|----------------|------------------|---------------|
| *Objaverse (3D dataset with mesh)* | | | | | | |
| Uni3D | 8.400 | 17.50 | 22.60 | point cloud | EVA02-E-14-plus | 1024 |
| CLIP[2] | 7.400 | 22.20 | 32.50 | point cloud | ViT-B-16 | 512 |
| CLIP[2] | 6.400 | 20.20 | 30.90 | 3DGS | ViT-B-16 | 512 |
| Uni3D | 2.300 | 8.100 | 12.00 | 3DGS location | EVA02-E-14-plus | 1024 |
| Uni3D* | 16.70 | 37.10 | 48.10 | point cloud | ViT-B-16 | 512 |
| Uni3D | 10.40 | 26.20 | 36.40 | 3DGS | ViT-B-16 | 512 |
| Uni3D* | 15.80 | 35.60 | 47.20 | 3DGS | ViT-B-16 | 512 |
| UniGS(Ours) | **21.00** | **39.80** | **53.50** | 3DGS | ViT-B-16 | 512 |
| *ABO (3D dataset with mesh)* | | | | | | |
| Uni3D | 5.700 | 19.19 | 29.49 | point cloud | EVA02-E-14-plus | 1024 |
| CLIP[2] | 7.090 | 24.34 | 38.94 | point cloud | ViT-B-16 | 512 |
| CLIP[2] | 7.650 | 23.92 | 37.83 | 3DGS | ViT-B-16 | 512 |
| Uni3D | 1.670 | 6.400 | 11.27 | 3DGS location | EVA02-E-14-plus | 1024 |
| Uni3D* | 10.29 | 29.21 | 43.67 | point cloud | ViT-B-16 | 512 |
| Uni3D | 8.070 | 25.31 | 37.41 | 3DGS | ViT-B-16 | 512 |
| Uni3D* | 10.85 | 29.76 | 42.98 | 3DGS | ViT-B-16 | 512 |
| UniGS(Ours) | **11.27** | **30.32** | **43.95** | 3DGS | ViT-B-16 | 512 |
| *MVImgNet (Multi-view dataset)* | | | | | | |
| CLIP[2] | 9.560 | 31.00 | 42.19 | point cloud | ViT-B-16 | 512 |
| CLIP[2] | 12.12 | 31.93 | 45.45 | 3DGS | ViT-B-16 | 512 |
| Uni3D | 1.400 | 7.690 | 11.42 | 3DGS location | EVA02-E-14-plus | 1024 |
| Uni3D* | 17.72 | 48.95 | 62.24 | point cloud | ViT-B-16 | 512 |
| Uni3D | 9.090 | 29.14 | 39.63 | 3DGS | ViT-B-16 | 512 |
| Uni3D* | 17.02 | 47.09 | 60.61 | 3DGS | ViT-B-16 | 512 |
| UniGS(Ours) | **18.65** | **53.38** | **66.90** | 3DGS | ViT-B-16 | 512 |

Table 1: **Top1, Top5 and Top10 Text-3D retrieval accuracy. Avg.:** the mean average retrieval accuracy. * denotes training from scratch.

**Implementation Details.** Following Section 3.5, we collect 146000, 7929, 3483, and 61871 objects, optimized for Objaverse (including Objaverse-LVIS for evaluation only ), ABO, MVImgNet, and SUN RGBD datasets, respectively. For the retrieval task, we randomly sample 1000 items to form the test set, and use the rest as training set. We train UniGS with a learning rate of 1e-4 for 15 epochs for the retrieval task and 50 epochs for the zero-shot classification and scene recognition tasks.

## 4.2 COMPARISONS TO STATE-OF-THE-ART

To demonstrate the effectiveness of our proposed method, we evaluate UniGS on the Text-3D retrieval, zero-shot classification, and scene understanding tasks and make comparisons with state-of-the-art methods (Zeng et al., 2023; Zhou et al., 2024). We further report the results of Uni3D (Zhou et al., 2024) trained with 3DGS for completeness. Note that CLIP[2] (Zeng et al., 2023) is retrained on our collected dataset for fair comparisons.

**Text-3D retrieval.** With the learned multi-modal representations of UniGS, we can naturally retrieve 3D shapes from the query text or images. Here, we focus on Text-driven retrieval, due to its importance for 3D asset search and downstream applications. Specifically, we retrieve 3D shapes from the test set by calculating the cosine similarity between the embedding of the query text prompt and 3D shapes in the gallery. We report Top1, Top5, and Top10 accuracy.

As shown in Table 1, UniGS outperforms the current state-of-the-art approaches across all datasets and improves the Top 1 retrieval accuracy of CLIP[2] and Uni3D on the Objaverse dataset by over

| Methods | Top1 | Avg. Top3 | Top 5 | Representation | Text-image Model | Embedding dim |
|---|---|---|---|---|---|---|
| *Objaverse-Lvis (3D dataset with mesh)* | | | | | | |
| Uni3D | 38.17 | 59.92 | 67.18 | point cloud | EVA02-E-14-plus | 1024 |
| CLIP$^2$ | 12.35 | 24.62 | 32.91 | point cloud | ViT-B-16 | 512 |
| CLIP$^2$ | 10.20 | 20.47 | 27.71 | 3DGS | ViT-B-16 | 512 |
| Uni3D | 5.130 | 11.20 | 13.27 | 3DGS location | EVA02-E-14-plus | 1024 |
| Uni3D* | 36.72 | 57.09 | 65.18 | point cloud | ViT- B-16 | 512 |
| Uni3D | 18.48 | 34.39 | 43.31 | 3DGS | ViT-B-16 | 512 |
| Uni3D* | 30.47 | 48.46 | 55.87 | 3DGS | ViT-B-16 | 512 |
| UniGS(Ours) | **38.57** | **60.54** | **68.96** | 3DGS | ViT-B-16 | 512 |
| *ABO (3D dataset with mesh)* | | | | | | |
| Uni3D | 68.94 | 90.49 | 94.15 | point cloud | EVA02-E-14-plus | 1024 |
| CLIP$^2$ | 22.58 | 43.83 | 54.56 | point cloud | ViT-B-16 | 512 |
| CLIP$^2$ | 19.06 | 38.48 | 48.71 | 3DGS | ViT-B-16 | 512 |
| Uni3D | 13.34 | 28.28 | 42.20 | 3DGS location | EVA02-E-14-plus | 1024 |
| Uni3D* | 37.60 | 59.68 | 70.22 | point cloud | ViT-B-16 | 512 |
| Uni3D | 27.57 | 50.60 | 63.69 | 3DGS | ViT-B-16 | 512 |
| Uni3D* | 37.79 | 61.08 | 69.04 | 3DGS | ViT-B-16 | 512 |
| UniGS(Ours) | **46.97** | **69.91** | **79.38** | 3DGS | ViT-B-16 | 512 |
| *MVImgNet (Multi-view dataset)* | | | | | | |
| CLIP$^2$ | 5.030 | 12.33 | 16.96 | point cloud | ViT-B-16 | 512 |
| CLIP$^2$ | 4.310 | 11.24 | 14.89 | 3DGS | ViT-B-16 | 512 |
| Uni3D | 3.680 | 10.72 | 14.92 | 3DGS location | EVA02-E-14-plus | 1024 |
| Uni3D* | 5.410 | 12.51 | 16.68 | point cloud | ViT-B-16 | 512 |
| Uni3D | 7.020 | 15.18 | 21.05 | 3DGS | ViT-B-16 | 512 |
| Uni3D* | 4.920 | 12.42 | 15.70 | 3DGS | ViT-B-16 | 512 |
| UniGS(Ours) | **7.650** | **16.96** | **22.48** | 3DGS | ViT-B-16 | 512 |

Table 2: **Zero-shot classification. Avg.:** the mean average classification accuracy. * denotes training from scratch.

| Method | Rep. | Avg. | Bed | Bsf. | Chair | Desk | Sofa | Table | Toilet | Btub. | Dresser | NSd. |
|---|---|---|---|---|---|---|---|---|---|---|---|---|
| Uni3D | point clouds | 11.04 | 56.74 | 14.33 | 19.58 | 25.03 | 22.93 | 15.98 | 24.62 | 24.00 | 0.000 | 1.690 |
| CLIP$^2$ | point clouds | 41.39 | 1.840 | 14.00 | 68.02 | 30.98 | 45.44 | 7.460 | 13.85 | 0.000 | 15.00 | 3.390 |
| CLIP$^2$ | 3DGS | 28.50 | 1.470 | 4.000 | 40.03 | 1.640 | 15.20 | 56.72 | 4.620 | 0.000 | 26.25 | 30.51 |
| Uni3D* | point clouds | 61.72 | 63.60 | **59.67** | 84.33 | **47.43** | **79.36** | **78.97** | 63.59 | 74.67 | 12.92 | 18.93 |
| Uni3D | 3DGS | 54.51 | 58.09 | 19.00 | 80.38 | 17.05 | 62.40 | 47.68 | 56.92 | 48.00 | 7.500 | 11.02 |
| Uni3D* | 3DGS | 56.67 | 74.63 | 28.00 | 83.89 | 28.36 | 50.88 | 54.31 | 7.690 | 20.00 | 27.50 | 19.49 |
| UniGS(Ours) | 3DGS | **69.64** | **81.62** | 32.00 | **87.46** | 17.38 | **79.36** | 68.74 | **93.85** | **96.00** | **35.00** | **36.44** |

Table 3: **Recognition on SUN RGBD (dataset with point clouds). Avg.:** the mean average Top1 accuracy across all categories. * denotes training from scratch.

13.6% and 5.2%, respectively. For reference we also include the evaluation results of Uni3D on their larger dataset (in gray).

**Zero-shot classification.** We evaluate the zero-shot classification performance of UniGS on Objaverse-Lvis, ABO, and MVImgNet without accessing their training sets. We reorganized ABO into 44 major categories, then skipped those with items fewer than 50, and formed 23 categories ultimately for classification evaluation. Similarly, we reorganized the Objaverse and MVImgNet datasets into 318 and 95 categories, respectively.

| Methods | Source | 3D points | Backbone | Avg. | | | Training Dataset | Representation |
|---------|--------|-----------|----------|------|------|------|------------------|----------------|
| | | | | Top1 | Top3 | Top 5 | | |
| *Additional zero-shot comparisons* | | | | | | | | |
| Uni3D | | | | 36.72 | 57.09 | 65.18 | | point cloud |
| Uni3D | | | | 30.47 | 48.46 | 55.87 | | 3DGS |
| TAMM | no Lvis | 1024 | EVA02-S | 22.70 | 38.83 | 47.13 | 100k | 3DGS |
| ReCon | | | | 23.40 | 41.41 | 48.95 | | 3DGS |
| UniGS(Ours) | | | | **38.57** | **60.57** | **68.96** | | 3DGS |

Table 4: **Summary of the experimental results on Objaverse-LVIS zero-shot classification. Avg.:** the mean average classification accuracy. All methods are trained from scratch.

| Training w/ point cloud | Initialization | | Avg. | | |
|-------------------------|----------------|------------------|------|------|------|
| | opacity | scale & rotation | Top1 | Top3 | Top 5 |
| ✗ | - | - | 46.97 | 69.91 | **79.38** |
| ✓ | 0 | 0 | 46.29 | 68.67 | 76.79 |
| ✓ | 0.4 | 0 | 48.49 | 70.06 | 77.55 |
| ✓ | 0.4 | 0.4 | **49.62** | **70.61** | 77.50 |

Table 5: **Zero-shot classification with point clouds on ABO.** Avg. denotes mean average classification accuracy. Results illustrate that properly converting point clouds into 3DGS format can improve performance.

As shown in Table 2, our UniGS significantly outperforms CLIP[2], improving performance by over 24% for both the Objaverse and ABO datasets. Due to the lack of real-world data in the pre-training dataset, the Top1 accuracy of MVImgNet is relatively low. However, UniGS outperforms all the baselines on all datasets, revealing the power of 3DGS representations and the effectiveness of UniGS.

**Scene recognition.** We leverage the SUN RGBD dataset (Song et al., 2015) as the scene data and classify objects into 37 categories following the setting of (Song et al., 2015). Since SUN RGBD only provides a single image for each scene, we load the point clouds as the initialization for 3D Gaussians and fix them during the entire 3DGS optimizations. Thus the 3DGS locations of SUN RGBD are equivalent to point clouds.

As shown in Table 3, our UniGS outperforms both CLIP[2] and Uni3D, improving performance by 28.25% and 7.92% respectively, and achieving an increase of 12.97% over directly modeling 3DGS. Moreover, the success of UniGS in SUN RGBD shows the robustness of 3DGS representation to the number of multi-view images.

**Further comparisons to additional zero-shot 3D object understanding methods.** As shown in the *Additional zero-shot comparisons* part of Table 4, we supplement extra comparisons to TAMM Zhang et al. (2024) and ReCon Qi et al. (2023) and UniGS significantly outperform SOTA methods, emphasizing the effectiveness of 3DGS representation and proposed Gaussian-aware Guidance.

### 4.3 GENERALIZATION TO POINT CLOUDS

To show the capability of UniGS to process vanilla point clouds, we conduct experiments where we convert point clouds into 3DGS to benefit the performance of modeling spatial information.As shown in Table 5, we do this by initializing the scale, opacity, and rotation with a default value and show that by including this data in training, UniGS can achieve better performance than trained using 3DGS only. This highlights UniGS's potential to inherit capabilities of the point cloud encoder and that it can directly be applied to 3D point clouds.

### 4.4 ABLATION STUDY.

**Ablation study on the proposed modules of UniGS.** To further study the power of 3DGS representation and the effectiveness of UniGS, we further ablate and evaluate UniGS in zero-shot classification on ABO. We summarize all the experiments and conduct an ablation study on whether to leverage the 3DGS feature, ViT pattern, Pretrained weight (Pre.), Parallel structure (Par.), and Cross attention (Cro.) between Parallel structures. Therefore, as shown in Table 6, we can analyze and conclude that

| ExP. | 3DGS | ViT | GAG | | | Avg. |
|---|---|---|---|---|---|---|
| | | | Pre. | Par. | Cro. | |
| 1 | ✗ | ✗ | ✗ | ✗ | ✗ | 22.62 |
| 2 | ✓ | ✗ | ✗ | ✗ | ✗ | 19.35 |
| 3 | ✓ | ✓ | ✗ | ✗ | ✗ | 38.38 |
| 4 | ✓ | ✓ | ✓ | ✗ | ✗ | 27.43 |
| 5 | ✓ | ✓ | ✓ | ✓ | ✗ | 37.58 |
| 6 | ✓ | ✓ | ✗ | ✓ | ✓ | 40.57 |
| 7 | ✓ | ✓ | ✓ | ✓ | ✓ | **46.97** |

Table 6: **Ablation study on the proposed modules.** Avg.: the mean average Top1 accuracy across all categories. GAG denotes our Gaussian-Aware Guidance.

| Data scaling up | | | |
|---|---|---|---|
| Data | 10k | 50k | 100k |
| Avg. | 27.67 | 43.06 | **46.97** |

| Model scaling up | | | |
|---|---|---|---|
| Model | UniGS-T | UniGS-S | UniGS-L |
| Avg. | 42.67 | 46.97 | **52.30** |

Table 7: **UniGS performance with scaling up. Avg.:** the mean average Top1 zero-shot classification accuracy on ABO.

- Comparisons between Exp1., Exp2., and Exp3. show that the convolution-based model, i.e. PointNet++, can not capture the explicit 3DGS feature while the ViT-based model, i.e. Uni3D, can successfully model feature relationships.

- Comparisons between Exp3., Exp4., and Exp7. show that loading pretrained weights has certain advantages, but without careful designs, it may not be possible to utilize the explicit features of 3DGS. On the contrary, training directly from scratch can better learn the feature information of 3DGS, revealing that incorrect model design can hinder subsequent learning due to prior knowledge.

- Comparisons between Exp3., Exp4., Exp5., and Exp7. show that dividing the model into parallel structures to process color and other 3DGS features separately can improve alignment of 3DGS, but it is not enough to achieve superior performance.

- Comparisons between Exp5. and Exp7. show the importance of cross attention to utilize the prior from point cloud encoding to guide 3DGS feature encoding.

- Comparisons between Exp5., Exp6., and Exp7. show that the pretrained weights are beneficial for the final performance and that the key to UniGS performance lies in the Gaussian-Aware Guidance that reduced the difficulty of overall 3DGS learning and enhances 3DGS understanding through the cross attentions.

**Scaling Up.** We next explore the effectiveness of scaling up training data and model type in Table 7. The performance under different data and model scales demonstrates that scaling up the training data and model size of UniGS can significantly improve the performance of 3D representation learning.

## 5 CONCLUSION

In this paper, we proposed UniGS, which for the first time includes 3DGS in cross-modal learning as the universal 3D representation supplementing shape and texture information. To this end, a parallel structure with cross-attention is proposed to avoid knowledge conflicts and builds knowledge connections via the attention mechanism. We demonstrate that our proposed UniGS achieves superior performance to state-of-the-art approaches and reveals the power and importance of 3DGS for 3D representation learning.

**Limitations:** Despite the robust and effective performance of UniGS for 3D representation learning and downstream applications, its current version lacks performance validation of out-door scenarios and 3D understanding ability with Large Language Model (LLM), resulting in insufficient performance improvement and validation for downstream tasks. Moreover, at least one image with a camera pose is required for the optimization of 3DGS, and how to further consider a camera-pose-free approach (e.g., Image-to-3DGS) or a dataset of pure point clouds while maintaining performance, is another exciting direction for future work.

**Acknowledgment:** This work is supported by National Key Research and Development Program of China(2024YFE0203100), National Natural Science Foundation of China (NSFC) under Grants No.62476293 and Nansha Key R&D Program under Grant No.2022ZD014. We thank the General Embodied AI Center of Sun Yat-sen University for support of this work.

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

## A  APPENDIX/SUPPLEMENTAL MATERIAL

The outline of the Appendix is as follows:

- More Implementation Details;
  - More Implementation Details of ensemble datasets;
  - More Implementation Details of training and evaluation;
- More ablation study on the quality of 3DGS;
  - Additional ablation study on objaverse zero-shot classification;
  - Additional ablation study on ABO zero-shot classification;
- More comparisons on large-scale data;
- More comparisons to state-of-the-art methods;
- More experiments with unfrozen fundamental Encoder;
- More experiments on generalization to 3DGS-driven methods;
- More evaluations on zero-shot image-driven retrieval;
  - Additional comparisons on zero-shot image-driven retrieval;
- Future work.
- Social impact;
- Discussion

## B  IMPLEMENTATION DETAILS

Here we provide more implementation details on the ensemble of the ObjaverseDeitke et al. (2023), ABOCollins et al. (2022), MVImgNetYu et al. (2023) and SUN RGBDSong et al. (2015) datasets; as well as the training and evaluation details on the Text-driven retrieval, Zero-shot classification, and scene recognition tasks.

### B.1  DETAILS OF ENSEMBLE DATASETS

Cap3DLuo et al. (2024) and InternLM-composer2Dong et al. (2024) is leveraged for object caption, which will be used for the Text-driven retrieval task. To better utilize the weights pre-trained on colored point clouds, the SH degree is set to 0 to ignore the view-dependent color effects, which also brings the additional advantage of reducing the computation and storage consumption. Moreover, the opacity reset interval is set to 501 to promote the stability of 3DGS. Note that the saving interval should not be a multiplier of the opacity reset interval, otherwise the retained results may become unstable. The optimization iteration for Objaverse, ABO, MVImgNet, and SUN RGBD datasets is set to 1500, 2000, 3000, and 500, respectively. All datasets can be successfully prepared on 6×RTX4090 GPU within 2 days, where 15 scenes can be optimized simultaneously on each GPU.

### B.2  DETAILS OF TRAINING AND EVALUATION

We leverage the activation function $tanh(\cdot)$ to convert the features of 3DGS to the range [-1,1] and set the batch size of training and evaluation to 24 and 80, respectively. In terms of time consumption, the whole training process on Objaverse costs 12.5 hours with 6×RTX4090 GPU, where UniGS is trained for 15 epochs on the Objaverse training set for the sufficient understanding and alignment of 3DGS representations, which will be used for Zero-shot classification and Text-driven retrieval in the inference stage.

**Zero-shot classification.**  After training 15 epochs on Objaverse, UniGS is directly evaluated on the entire Objaverse-Lvis, ABO, and MVimgnet datasets. We reorganize each dataset to accelerate the entire evaluation, where Objaverse-Lvis, ABO, and MVImgNet are reorganized into 315, 23, and 95 categories, respectively.

**Text-driven retrieval.**  In Text-driven retrieval, ABO and MVImgNet will be split into training and testing sets, where the testing sets of Objaverse, ABO, and MVImgNet contain 1000, 433, and 1450

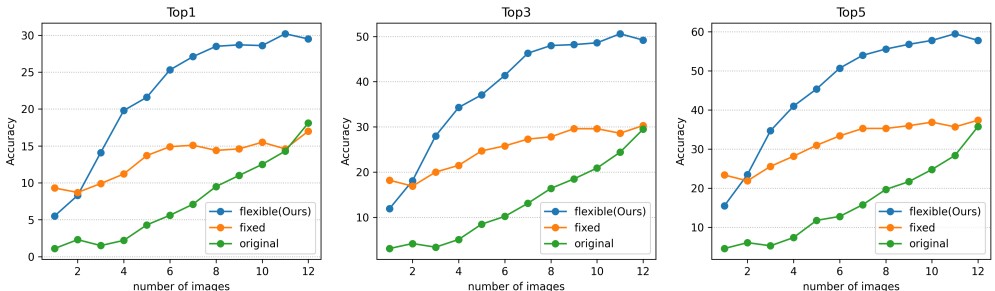

Figure 4: **Additional ablation study of the quality of 3DGS on the Text-driven retrieval task.** The accuracy of Text-driven retrieval on Objaverse under three optimization pipelines.

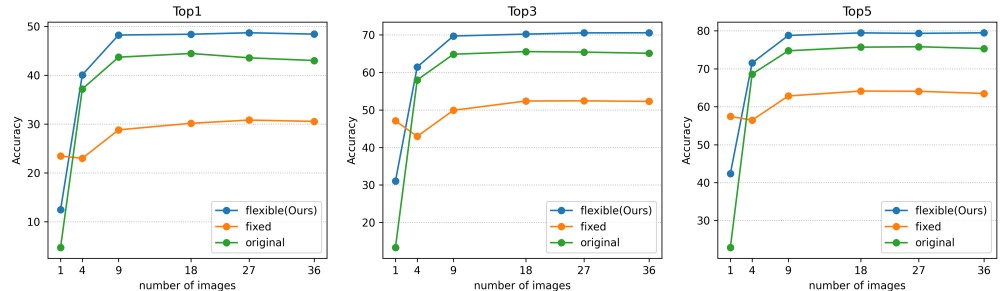

Figure 5: **Additional ablation study of the quality of 3DGS on the Zero-shot classification task.** The accuracy of Zero-shot classification on ABO under three optimization pipelines.

items, respectively. UniGS will be further fine-tuned for 50 epochs on the training set to alleviate the impact of the text domain across different datasets. Next, 3DGS encoded by UniGS is used to compute similarity and calculate Top$k$ accuracy across texts of all the items in the testing set.

**Scene recognition.** As for the scene recognition task on the SUN RGBD dataset, UniGS follows the basic evaluation pattern to directly train 50 epochs on the training set and finally be evaluated on the testing set.

## C ADDITIONAL ABLATION STUDY OF THE QUALITY OF 3DGS

As shown in Fig. 4 and Fig. 5, we considered two common optimization settings of 3DGS for the ablation of the number of images: (1) flexible (ours): load surface points as initialization with flexibility of 3DGS location, (2) original: the vanilla optimization from 3DGS.

As shown in the results in Fig. 4, with the increasing number of images, the overall accuracy generally shows an upward trend. Our "flexible" pipeline exhibits stronger robustness and better reconstruction capability to the number of input images. Notably, when the number of images is halved, the overall accuracy can be maintained within 10% of the optimal level. Moreover, as shown in the results in Fig. 5 of the rebuttal PDF, our additional ablation study on the ABO dataset reveals that 3DGS only requires a small number of images to achieve commendable results. More specifically, on simpler reconstruction tasks like ABO, our UniGS can maintain its performance even when the image count is drastically reduced from 36 to just 4.

## D FURTHER COMPARISONS ON LARGE-SCALE DATA

As shown in Table 8, we supplement comparisons between UniGS and Uni3D when scaling up to the dataset used in Uni3D, which is a combination of Objaverse and ABO. As shown in the *Large-scale training* part in Table, UniGS benefits from scaling up of the dataset and still outperforms Uni3D when considering the larger scale setting, clearly demonstrating the benefit of 3DGS over point clouds. Additional comparisons to the official state-of-the-art methods with 10000 3D points can be found in Appendix E.

Table 8: **Experimental results of large scale training on Objaverse-LVIS zero-shot classification. Avg.:** the mean average classification accuracy. All methods are trained from scratch.

| Methods | Source | 3D points | Backbone | Top1 | Avg. Top3 | Top 5 | Training Dataset | Representation |
|---------|--------|-----------|----------|------|-----------|-------|------------------|---------------|
| *Large-scale training* | | | | | | | | |
| Uni3D | with Lvis | 1024 | EVA02-S | 46.31 | 72.62 | 79.78 | 800k | 3DGS |
| UniGS(Ours) | | | | **49.95** | **75.60** | **82.38** | | |

Table 9: **Comparisons to state-of-the-art methods with the same data on Objaverse-LVIS zero-shot classification. Avg.:** the mean average classification accuracy.

| Methods | Source | Backbone | Top1 | Avg. Top3 | Top 5 | Dataset train | test | Representation |
|---------|--------|----------|------|-----------|-------|---------------|------|---------------|
| *10000 3D points with official model* | | | | | | | | |
| TAMM | | Point-BERT | 50.70 | 73.20 | 80.60 | 800k | 46k | point clouds |
| ReCon++-B | | ViT-bigG | 53.20 | 75.30 | 81.50 | 800k | 46k | point clouds |
| Uni3D-S | with Lvis | EVA02-S | 50.34 | 72.70 | 79.81 | 800k | 46k | point clouds |
| Uni3D-S | | EVA02-S | 49.87 | 72.39 | 79.70 | 800k | 6k | point clouds |
| UniGS-S(Ours) | | EVA02-S | 51.22 | 73.64 | 80.88 | 46k | 6k | 3DGS |

## E  FURTHER COMPARISONS TO STATE-OF-THE-ART METHODS

We supplement extra experiments on representations with 10000 3D points and evaluate UniGS on a no-bias mini testing set following setting of the "Ensembled (with LVIS)" from Uni3D. As shown in Table 9, UniGS benefits from the increasing of 3D points, achieving similar Top 1 accuracy compared with Uni3D, TAMM and ReCon++ (regardless of backbones). Note that TAMM and ReCon++ use larger model parameters than UniGS.

Therefore, as shown in Table 4 and Table 9, with the dataset scaling up and the increasing of 3D poitns, UniGS have great potential to show superior performance over common point clouds, outperforming at the same data.

Moreover, UniGS proposed to leverage 3DGS as 3D representation and utilize Gaussian-Aware Guidance for better understanding, which can be applied at existing 3D understanding work with point clouds for improvement. Since we leverage Uni3D as our fundamental encoder, experiments will focus on the comparisons with Uni3D in the original paper to show the effectiveness of 3DGS representation and our Gaussian-Aware Guidance module.

## F  ADDITIONAL ABLATION STUDY WITH UNFROZEN REPRESENTATION

We freeze the Uni3D encoder as our UniGS only leverages spatial information of point clouds for explicit feature learning. Therefore, we opt to freeze the Uni3D encoder, utilizing it solely for guidance, which leads to relationship modeling and feature understanding of the 3DGS explicit features, instead of directly training our point encoder on 3DGS.

Moreover, we provide additional experiments with unfrozen fundamental encoder in Table 10. Despite the doubled demand for computing resources, UniGS with an unfrozen fundamental encoder still achieves comparable performance.

## G  GENERALIZATION TO 3DGS-DRIVEN METHODS

We provide more comprehensive results of the robustness to 3DGS-derived methods with UniGS trained on 2DGS representation. As shown in the Table above, UniGS pre-trained on 2DGS achieve similar performance compared with UniGS pre-trained on 3DGS, illustrating the power of our proposed Gaussian-Aware Guidance and the ability to migrate to other 3DGS-derived methods.

Table 10: **Additional comparisons to state-of-the-art methods on Zero-shot classification.** Avg. denotes the mean average classification accuracy. * denotes training from scratch and †denotes unfreezing the fundamental encoder.

| Methods | Ufrozen fundamental encoder | *Objaverse* Avg. | | | *ABO* Avg. | | |
|---|---|---|---|---|---|---|---|
| | | Top1 | Top3 | Top 5 | Top1 | Top3 | Top 5 |
| TAMM* | - | 22.70 | 38.83 | 47.13 | 35.44 | 54.96 | 63.40 |
| ReCon* | - | 23.40 | 41.41 | 48.95 | 34.29 | 55.14 | 67.69 |
| Uni3D* | - | 30.47 | 48.46 | 55.87 | 37.79 | 61.08 | 69.04 |
| UniGS(Ours)* | ✗ | 38.57 | 60.54 | 68.96 | 46.97 | 69.91 | **79.38** |
| UniGS(Ours)† | ✓ | **38.74** | **62.89** | **71.88** | **47.53** | **70.49** | 78.60 |

Table 11: **Zero-shot classification on Objaverse with other 3DGS-driven methods.** Avg. denotes mean average classification accuracy. Results illustrate the ability of UniGS to migrate to other 3DGS-driven methods. 2DGS denotes "*2D Gaussian Splatting for Geometrically Accurate Radiance Fields*".

| Representation Training & Testing | Avg. | | |
|---|---|---|---|
| | Top1 | Top3 | Top 5 |
| 3DGS | 38.57 | 60.54 | 68.96 |
| **2DGS** | 38.11 (-0.46) | 61.28 (+0.74) | 70.32 (+1.36) |

# H  ZERO-SHOT IMAGE-DRIVEN RETRIEVAL

As shown in Table 12, we further evaluate UniGS with Zero-shot Image-driven retrieval in batch, which reveals the alignment between 3DGS and the image domain. Experiment results on ABO and MVImgNet in Table 12 demonstrate the power of 3DGS in Image-3D alignment.

# I  FUTURE WORK

To further validate the scaling ability of UniGS for better performance and 3D understanding. The next step of UniGS is to conduct it on large 3DGS datasets with more 3D points and further explore model architectures for language-image-3D alignment. We will establish more pipelines that support the conversion of various datasets into 3DGS and further expand the scale of the 3DGS datasets. Another interesting future direction will be the model architecture exploration, where we plan to further explore the connection between fundamental encoder and advanced encoder, and attempt to support training on point cloud datasets for larger datasets and benchmarks.

# J  SOCIAL IMPACT

While there is a wide range of application domains where 3D representation learning will be beneficial, such as in autonomous driving, augmented/virtual reality, and embodied AI, there are also potentially negative application scenarios. For instance, these approaches could be used in malicious contexts to obtain private image data through splatting with a specific decoder or for surveillance purposes. Consequently, UniGS is released as a research tool to benefit the academic field only.

# K  DISCUSSION

## K.1  COMPARED TO NERF-BASED APPROACHES

As shown in Table 13, we conduct additional comparisons with NeRF-based approaches and UniGS outperforms nerf2clip(Ballerini et al., 2024) and nf2vec(Ramirez et al., 2023) with 9.93% and 6.64%, demonstrating significant improvement over NeRF-based approaches on cross-modalities learning.

| Methods | Top1 | Avg. Top3 | Top 5 | Representation | Text-image Model | Embedding dim |
|---------|------|-----------|-------|----------------|------------------|---------------|
| ***Objaverse-Lvis*** | | | | | | |
| CLIP[2] | 28.83 | 51.43 | 63.57 | point cloud | | |
| CLIP[2] | 27.99 | 50.30 | 62.76 | 3DGS | | |
| Uni3D* | 39.65 | 60.72 | 70.51 | point cloud | ViT-B-16 | 512 |
| Uni3D | 35.82 | 58.35 | 69.63 | 3DGS | | |
| Uni3D* | 34.74 | 56.70 | 67.12 | 3DGS | | |
| UniGS(Ours) | **41.78** | **62.50** | **72.24** | 3DGS | ViT-B-16 | 512 |
| ***ABO*** | | | | | | |
| CLIP[2] | 15.29 | 31.74 | 42.74 | point cloud | | |
| CLIP[2] | 13.80 | 29.60 | 40.85 | 3DGS | | |
| Uni3D* | 18.25 | 35.26 | 45.29 | point cloud | ViT-B-16 | 512 |
| Uni3D | 21.14 | 38.88 | 49.38 | 3DGS | | |
| Uni3D* | 25.30 | 45.69 | 57.51 | 3DGS | | |
| UniGS(Ours) | **26.69** | **46.26** | **56.72** | 3DGS | ViT-B-16 | 512 |
| ***MVImgNet*** | | | | | | |
| CLIP[2] | 6.900 | 17.65 | 26.16 | point cloud | | |
| CLIP[2] | 2.160 | 6.330 | 10.12 | 3DGS | | |
| Uni3D* | 6.380 | 16.65 | 25.50 | point cloud | ViT-B-16 | 512 |
| Uni3D | 1.410 | 4.260 | 6.840 | 3DGS | | |
| Uni3D* | 7.940 | 18.86 | 27.20 | 3DGS | | |
| UniGS(Ours) | **10.55** | **23.75** | **33.15** | 3DGS | ViT-B-16 | 512 |

Table 12: **Zero-shot Image-driven retrieval. Avg.:** the mean average retrieval accuracy. * denotes training from scratch.

| Methods | 3D Representation | Avg.(%)↑ |
|---------|-------------------|----------|
| ***ShapeNetRender*** | | |
| CLIP(1 view) | — | 73.60 |
| CLIP(16 view) | — | 82.40 |
| nerf2clip | NeRF | 84.00 |
| nf2vec | NeRF | 87.30 |
| Uni3D | 3DGS location | 88.96 |
| UniGS(Ours) | 3DGS | **93.94** |

Table 13: **Zero-shot classification on ShapeNetRender. Avg.:** the mean average Top1 classification accuracy. Uni3D and UniGS is trained for 15 epoch on ShapeNetRender.

### K.2 UNIGS VS UNI3D

As shown in Fig. 6, we highlight the difference between UniGS and Uni3D. Our UniGS is also capable of utilizing a single model to unify 3D representations from different models, with better performance with 3DGS representation and proposed Gaussian-Aware Guidance. Specifically, when using point clouds as a unified 3D representation, the main challenge is the divergence between the 3D representation and other modalities. In contrast, UniGS leverages 3DGS as the 3D representation, which effectively reconstructs the 3D target object as well as provides efficient correspondences between 3D and 2D images.

Moreover, we conducted additional experiments to fine-tune UniGS using a pure 3DGS dataset. As shown in Table 14, UniGS outperforms Uni3D when augmented with point clouds under the same settings as Uni3D. It also achieves higher Top-1 and Top-3 accuracy after fine-tuning on the pure

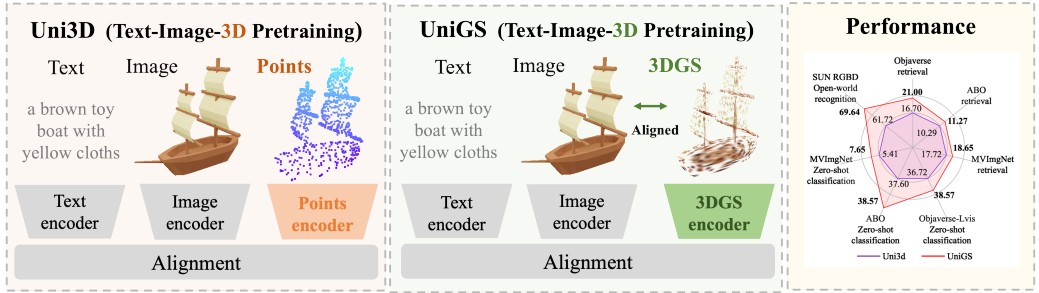

Figure 6: **Left:** Uni3D, using 3D point clouds for text-image-3D pre-training. **Middle:** our UniGS, leveraging 3DGS as the 3D representation with better alignment with image modality. **Right:** our UniGS learns a more general and stronger multi-modal representation.

Table 14: **Comparison results with 10000 points dataset on Objaverse-Lvis zero-shot classification.** † denotes fine-tuning on 3DGS datasets.

| Methods | Backbone | Top1 | Avg. Top3 | Top 5 | Representation | Augment w. point clouds |
|---------|----------|------|-----------|-------|----------------|-------------------------|
| ***10000 3D points*** | | | | | | |
| Uni3D | | 50.34 | 72.70 | 79.81 | point clouds | — |
| UniGS | EVA02-S-patch14 | 52.44 | 75.37 | **82.71** | 3DGS | ✓ |
| UniGS† | | **53.16** | **75.59** | 82.14 | 3DGS | ✗ |

3DGS dataset. Experimental results in Table 2 further demonstrate that Uni3D is not fully compatible with both point clouds and 3DGS. However, with our proposed Gaussian-Aware Guidance, UniGS exhibits the ability to effectively understand objects in both point clouds and 3DGS, achieving superior results after fine-tuning.

### K.3 RUNTIME ANALYSIS

Table 15: **Comparisons of forward computational cost on Objaverse-Lvis.**

| Methods | FLOPs(G) ↓ | Time(ms)↓ | Top 1 Avg. |
|---------|-----------|-----------|------------|
| CLIP[2] | 22.49 | 232 | 10.20 |
| TAMM | 22.49 | 233 | 22.70 |
| Uni3D | 47.85 | 113 | 30.47 |
| UniGS(Ours) | 98.17 | 233 | **38.57** |

As shown in the Table 15, we further evaluate the FLOPs and runtime of UniGS and compare them with state-of-the-art approaches. With a slight increase in runtime, UniGS achieves significant improvement over CLIP[2], TAMM, and Uni3D on Objaverse-Lvis zero-shot classification.

Moreover, as shown in the Table 16 and Fig. 7, we present an additional ablation study of UniGS modules on FLOPs in Fig. 7. Specifically, this helps us understand the difference as 76.5% of the total FLOPs (73.38G) is due to the CNN layers of the 3D Encoder to extract 3D spatial features.

Fortunately, much progress is being made in compressing models(Yang et al., 2024) and 3D representations(Fan et al., 2023), and we expect these advances to facilitate the development of 3D understanding with 3DGS representation.

### K.4 VISUAL COMPARISONS

As shown in Fig. 8, Uni3D may mistakenly retrieve another similar object due to the similarity in point cloud structure. In contrast, UniGS demonstrates a superior 3D understanding of object color,

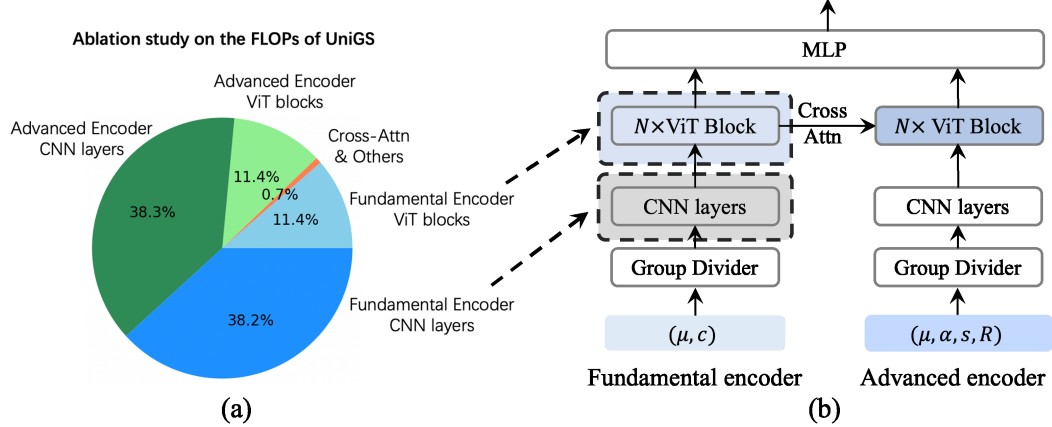

(a)                      (b)

Figure 7: **Additional ablation study on the FLOPs of UniGS.** (a) FLOPs of each module, (b) 3D Encoder of UniGS. 76.5% of FLOPs are due to the CNN layers of the fundamental and the advanced encoder to extract 3D spatial features, while only 23.5% of the FLOPs is spent for 3D understanding.

| Fundamental Encoder | | Advanced Encoder | | Cross-Attn | Others | FLOPs |
|---|---|---|---|---|---|---|
| CNN layers | ViT blocks | CNN layers | ViT blocks | | | |
| ✓ | ✗ | ✗ | ✗ | ✗ | ✗ | 36.67 |
| ✓ | ✓ | ✗ | ✗ | ✗ | ✗ | 47.60 |
| ✓ | ✓ | ✓ | ✗ | ✗ | ✗ | 84.31 |
| ✓ | ✓ | ✓ | ✓ | ✗ | ✗ | 95.24 |
| ✓ | ✓ | ✓ | ✓ | ✓ | ✗ | 95.43 |
| ✓ | ✓ | ✓ | ✓ | ✓ | ✓ | 95.94 |

Table 16: **Ablation study on the FLOPs of UniGS modules.** CNN encoder denotes the CNN layers to extract spatial information from 3D representation into features, and ViT blocks denotes the Transformer blocks understanding object from extracted features. Cross-Attn denotes the Cross-attention layers between Fundamental and Advanced Encoder.

shape, and texture with 3DGS representation and proposed Gaussian-Aware Guidance, resulting in better Image-to-3D retrieval.

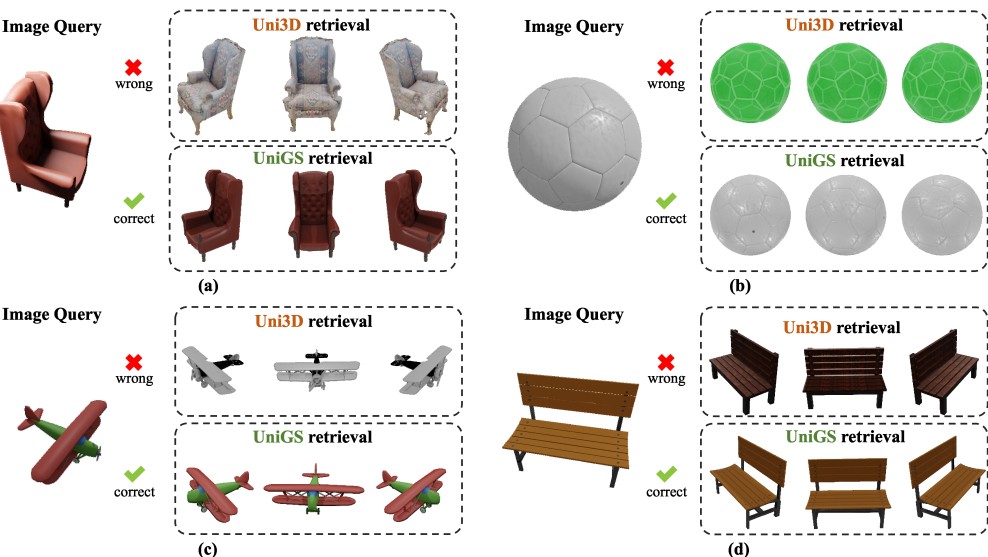

Figure 8: **Visual comparisons to Uni3D on Image-to-3D retrieval.** With 3D Gaussian Splatting representation capturing image details and a powerful Gaussian-Aware Guidance module, UniGS outperforms Uni3D in 3D understanding of object color, shape, and texture.

