# OpenReview forum: "UniGS: Unified Language-Image-3D Pretraining with Gaussian Splatting"
_ICLR.cc/2025/Conference — ICLR 2025 Poster_

### Official Review · Reviewer_SAh2 · 2024-10-28

**Soundness:** 3
**Presentation:** 3
**Contribution:** 3
**Rating:** 6
**Confidence:** 4

**Summary:**

This paper argues that point clouds, as 3D representations, cannot fully capture the intricacies of the 3D world. To address this, it proposes the use of 3D Gaussian primitives instead of point clouds. The authors introduce a pretraining strategy to align the 3D Gaussian features with those of a pretrained vision-language model, establishing a shared visual and textual space through extensive real-world image-text pairs. Additionally, they propose a Gaussian-Aware Guidance module that leverages priors from pretrained point cloud encoders to guide the learning of Gaussian features, enhancing 3D understanding. The proposed approach achieves state-of-the-art performance across various challenging datasets.

**Strengths:**

1. Introduction of 3D Gaussian Primitives: It proposes a novel representation using 3D Gaussian primitives, offering a potential alternative to point clouds for improved 3D modeling.
2. Alignment with Vision-Language Models via Pretraining: The pretraining strategy aligns 3D Gaussian features with a pretrained vision-language model, enabling a shared visual-textual space through large-scale real-world image-text pairs.
3. Gaussian-Aware Guidance Module: The proposed module leverages pretrained point cloud encoders to guide the learning process, helping the 3D Gaussian features improve their understanding and representation capabilities.
4. State-of-the-Art Performance: The method demonstrates superior results, achieving state-of-the-art performance across some of the datasets against point cloud-based counterparts.

**Weaknesses:**

This paper has several weaknesses that should be addressed:

1. Lack of Comparison with Related Works: While the authors contend that 3D Gaussian primitives (3DGS) offer a superior 3D representation, the paper does not provide comparisons with other relevant works, such as [1], [2], [3], and [4]. These works utilize Neural Radiance Fields (NeRF), which offer similar advantages for 3D perception tasks, as mentioned in Lines 63-70. Including such comparisons would strengthen the argument.
2. Insufficient Clarity in the Implementation Details: The paper lacks clear explanations regarding the methodology, leaving certain aspects ambiguous. This raises multiple questions that require further clarification, as outlined in Questions 3 and 4.
3. Unsubstantiated Hypothesis: Without addressing the aforementioned issues, the core hypothesis that 3D Gaussian primitives are a better representation than point clouds remains unconvincing. A thorough comparison and clearer implementation details are necessary to support this claim effectively.
4. Improving these weaknesses could strengthen the paper’s claims, which may lead me to consider raising its rating.

**Questions:**

1. Clarification on NeRF-based Methods: Could you clarify why NeRF-based methods were not discussed in this work? Are there any specific disadvantages that prevented you from considering them as alternatives to 3DGS?

2. Evaluation of Computational Time: Have you evaluated the runtime of your method? While 3D Gaussian primitives may offer several advantages, they require an additional optimization process that point clouds do not, as point clouds represent raw data directly from 3D sensors. This suggests that 3DGS involves additional time for optimization, which is not discussed in the paper. Could you provide more details on this aspect?

3. Initialization with Point Clouds: For clarification, did you use sample raw point clouds from meshes to initialize the Gaussian primitives for the experiments (for ABO and Objaverse) presented in your main results? If so, could you explain why, in some datasets (such as ABO in Table 2), the performance of 3DGS falls short compared to point cloud methods?

4. Clarification on Baseline Performance and Experimental Settings: Could you elaborate on how the performance of the baseline models was obtained? Additionally, how does the experimental setting of the results reported in your paper (Table 2, Lines 328-329, showing 38.17% top-1 accuracy) differ from the Objaverse-LVIS zero-shot classification performance reported in Uni3D[5] (Table 1)? A detailed comparison would clarify any discrepancies.

**References:**

[1] Jeong, et al. (2022). Perfception: Perception using radiance fields. Advances in Neural Information Processing Systems, 35, 26105-26121.

[2] Hu, et al. (2023). Nerf-rpn: A general framework for object detection in nerfs. In *Proceedings of the IEEE/CVF Conference on Computer Vision and Pattern Recognition,* (pp. 23528-23538)

[3] Li, et al. (2024). GP-NeRF: Generalized Perception NeRF for Context-Aware 3D Scene Understanding. In Proceedings of the IEEE/CVF Conference on Computer Vision and Pattern Recognition (pp. 21708-21718).

[4] Ballerini, et al. (2024). Connecting NeRFs Images and Text. In Proceedings of the IEEE/CVF Conference on Computer Vision and Pattern Recognition Workshop (pp. 866-876).

[5] Zhou, et al. (2024). Uni3D: Exploring Unified 3D Representation at Scale. In The Twelfth International Conference on Learning Representations.

---

> ### Author Response · Authors · 2024-11-20
>
> Thank you for your detailed and careful feedback on our paper. We are pleased that you recognize the contributions of UniGS on the
> introduction of 3D Gaussian primitives for 3D-oriented topics. We are also glad to hear that you appreciate the novelty and effectiveness of UniGS. We will address your concerns point by point:
>
> >
> >* __W1, Q1(Comparisons to NeRF-based approaches)__ : "the paper does not provide comparisons with other relevant works ... These works utilize Neural Radiance Fields (NeRF)"
> >
>
> | Methods         | 3D Representation   | Avg.(%) ↑      |
> |------------------|---------------------|----------------|
> | **ShapeNetRender** |                     |                |
> | CLIP (1 view)    | ---                 | 73.60          |
> | CLIP (16 view)   | ---                 | 82.40          |
> | nerf2clip        | NeRF                | 84.00          |
> | nf2vec           | NeRF                | 87.30          |
> | Uni3D            | 3DGS location       | 88.96          |
> | **UniGS (Ours)** | 3DGS                | **93.94**      |
>
> **Table 1**: Zero-shot classification on ShapeNetRender. Avg.: the mean average Top1 classification accuracy. Uni3D and UniGS are trained for 15 epochs on ShapeNetRender.
>
>
> __`Comparisons to NeRF-based approaches`__: Thank you for the suggestion. We have now conducted additional comparisons with NeRF-based approaches. As shown in Table 1, UniGS outperforms nerf2clip[1] and nf2vec[2] with 9.93\% and 6.64\%, demonstrating significant improvement over NeRF-based approaches on cross-modalities learning.
>
> __`Why not compared with NeRF-based approaches in the main paper`__: When utilizing point clouds for 3D representation, a significant challenge arises from the discrepancy between the 3D representation and other modalities. While NeRF can achieve pixel alignment with provided images, it has several drawbacks: its implicit representation is not as advantageous for 3D tasks as the explicit representation of 3DGS. Additionally, NeRF optimization is notably slow and demands a substantial number of viewpoints. In contrast, as illustrated in Table 5 of the main paper, 3DGS showcases potential compatibility with point clouds, offering promise for practical applications in real-world scenarios.
>
> __`Summary`__: Thank you very much for your suggestions. As the additional experiments demonstrated, our framework achieves significant improvements with 3DGS representation over NeRF. We have incorporated the NeRF experiments and discussions into the paper to strengthen the persuasiveness and completeness of our work.
>
> [1] Connecting NeRFs, Images, and Text
>
> [2] Deep learning on 3D neural fields.

---

> ### Author Response · Authors · 2024-11-20
>
> >
> >* __Q2(Evaluation of Computational Time)__ : "3DGS ... requires an additional optimization process that point clouds do not, as point clouds represent raw data directly from 3D sensors"
> >
> | Methods       | FLOPs(G) ↓ | Time(ms) ↓ | Top 1 Avg.   |
> |---------------|------------|------------|--------------|
> | CLIP²         | 22.49      | 232        | 10.20        |
> | TAMM          | 22.49      | 233        | 22.70        |
> | Uni3D         | 47.85      | 113        | 30.47        |
> | UniGS(Ours)   | 98.17      | 233        | **38.57**    |
>
> **Table 2**: Comparisons of forward computational cost on Objaverse-Lvis.
>
>
> __`Runtime analysis`__:Thank you for the comment. As shown in the Table 2, we further evaluate the FLOPs and runtime of UniGS and compare them with state-of-the-art approaches. With a slight increase in runtime, UniGS achieves significant improvement over ${CLIP}^{2}$, TAMM, and Uni3D on Objaverse-Lvis zero-shot classification.
>
> | Fundamental Encoder |            | Advanced Encoder |            | Cross-Attn | Others | FLOPs  |
> |---------------------|------------|------------------|------------|------------|--------|--------|
> | CNN layers          | ViT blocks| CNN layers       | ViT blocks |            |        |        |
> | ✓                   |          |                 |           |           |       | 36.67  |
> | ✓                   | ✓         |                 |           |           |       | 47.60  |
> | ✓                   | ✓         | ✓                |           |           |       | 84.31  |
> | ✓                   | ✓         | ✓                | ✓          |           |       | 95.24  |
> | ✓                   | ✓         | ✓                | ✓          | ✓          |       | 95.43  |
> | ✓                   | ✓         | ✓                | ✓          | ✓          | ✓      | 95.94  |
>
> **Table 3**: Ablation study on the FLOPs of UniGS modules. CNN encoder denotes the CNN layers to extract spatial information from 3D representation into features, and ViT blocks denotes the Transformer blocks understanding objects from extracted features. Cross-Attn denotes the Cross-attention layers between Fundamental and Advanced Encoder.
>
> Moreover, as shown in the Table 3 and Figure 7 in Sec. J of the appendix, we further conduct an in-depth evaluation of the FLOPs for our UniGS framework, which provides a better understanding of the increased computational cost. Specifically, __76.5\% of the total FLOPs (73.38G) is costed in the CNN layers of 3D Encoder to extract 3D spatial features__ for understanding.
> This is indeed a limitation of our method's general application. Fortunately, much progress is being made in compressing models [1] and 3D representations [2], and we expect these advances to facilitate the development of 3D understanding with 3DGS representation.
>
>
> __`Time analysis of 3DGS optimization`__: Indeed, 3DGS involves additional time for optimization and the details of the required 3DGS optimization cost are discussed in Line at line 740 in the main paper.
> Specifically, preparing 800k 3DGS objects with 1024 3D Gaussian kernels takes a week to prepare, while 800k 3DGS objects with 10000 3D Gaussian kernels takes about 12 days.
> The preparation of 3DGS datasets demands significant efforts in terms of time and computational power. We will therefore make all prepared datasets publicly available to the community to support further advancements in 3DGS representation learning.
>
> Moreover, leveraging image-to-3DGS approaches for dataset preparation is another promising step. For example, GS-LRM [3] and PF-LRM [4] takes only 0.23 and 1.3 seconds from 2-4 posed sparse images to generate 3DGS, respectively. Although recent advances have been made on image-to-3DGS [3,4], they are unfortunately still not open-sourced now. Given the popularity of 3DGS, we expect these representations to only become more and more efficient to construct.
>
> [1] T3DNet: Compressing Point Cloud Models for Lightweight 3D Recognition
>
> [2] LightGaussian: Unbounded 3D Gaussian Compression with 15x Reduction and 200+ FPS
>
> [3] GS-LRM: Large Reconstruction Model for 3D Gaussian Splatting
>
> [4] PF-LRM: Pose-Free Large Reconstruction Model for Joint Pose and Shape Prediction

---

> > ### Comment · Reviewer_SAh2 · 2024-11-22
> >
> > Thank you for your insights. I wanted to bring to your attention a relevant open-source project: [OpenLRM](https://github.com/3DTopia/OpenLRM), which I believe is also trained on MVImgNet. I agree with your observation that feedforward Gaussian Splatting methods significantly accelerate the process of generating Gaussian primitives. I thought this might be a helpful reference for further exploration.

---

> ### Author Response · Authors · 2024-11-20
>
> >
> >* __W2,Q3(Initialization with Point Clouds)__ : "in some datasets (such as ABO in Table 2), the performance of 3DGS falls short compared to point cloud methods"
> >
>
> __`Why initialized with Point Clouds`__:
> As shown in Figure 4 and 5 in Sec. B of the appendix, we considered three common optimization settings of 3DGS for the ablation of the initialization of 3DGS: (1) flexible (ours): load surface points as initialization with flexibility of 3DGS location, (2) fixed: load surface points as initialization with fixed 3DGS location, (3) original: the vanilla optimization from 3DGS.
>
> As shown in the results in Figure 4 and 5 in Sec. B of the appendix, pipelines that load surface points for initialization achieve better results over the vanilla optimization from 3DGS. Moreover, our "flexible" pipeline shows significant improvement with flexibility of 3DGS instead of fixing the location of 3DGS. Therefore, UniGS leverages the "flexible" pipeline for data preparation, which means __3DGS will learn the most representative locations of an object, rather than manually selected fixed points on the surface of the object__.
>
>
> __`Why Uni3D with 3DGS falls short compared to point cloud`__: Since the spatial information of 3DGS is not as regular as that of point clouds, which are typically located on the surface, it presents challenges to learning 3DGS features for 3D understanding. However, the 3D Encoder of Uni3D is designed for point clouds, which is not suitable for 3DGS and may deteriorate performance.
>
> Therefore, we proposed Gaussian-Aware Guidance for UniGS to better understand the 3DGS features with the spatial query from the fundamental encoder. As shown in the Table 2 and 6 of the main paper, our UniGS without the spatial query from cross attention gets only 37.58\% Top1 accuracy on ABO, which is close to the 37.79\% Top 1 accuracy that we train Uni3D from scratch on 3DGS, demonstrating the effectiveness of our proposed Gaussian-Aware Guidance.
>
> >
> >* __W2, Q4(Clarification on Baseline Performance and Experimental Settings)__ : "how does the experimental setting of the results reported in your paper... differ from the Objaverse-LVIS zero-shot classification performance reported in Uni3D"
> >
>
> | Methods | Backbone            | Top1   | Top3   | Top5   | Representation  |
> |---------|---------------------|--------|--------|--------|-----------------|
> | **10000 3D points**           |                     |        |        |        |                 |
> | Uni3D   | EVA02-S-patch14     | 50.34  | 72.70  | 79.81  | point clouds    |
> | UniGS   | EVA02-S-patch14     | **53.16** | **75.59** | **82.14** | 3DGS            |
>
> **Table 4**: Comparison results with 10000 points dataset on Objaverse-Lvis zero-shot classification.
>
> __`Comparisons under the setting of Uni3D`__: Thank you for the suggestion. For comprehensive comparisons, we conduct further comparisons with the same setting of Uni3D and reformulate experiment results in Table 4. __Under the same setting of Uni3D, UniGS still outperforms Uni3D on Objaverse-Lvis zero-shot classification__.
>
> __`Performance of the baseline models`__: UniGS outperform Uni3D under a fair setting of representing object with 1024 3D points and training with CLIP ViT-B-16. In the following, we briefly elaborate on the performance gap in Top 1 performance:
>
> (1) __model type__: reported Top 1 performance of Uni3D uses the 1 billion parameter model (Uni3d-g) for better results while we only use the small version for consideration of over-fitting. The Top 1 accuracy of Uni3d-s on Objaverse-LVIS reduces to 50.34.
>
> (2) __training mode__: we conduct experiments on Zero-shot classification, which means the test set will not be used in training, corresponding to the not ensemble mode of Uni3D. When not using the ensemble mode and leveraging the smaller model, the Top 1 accuracy of Uni3D-S on Objaverse-LVIS comes to 44.81.
>
> (3) __the number of points__: we set the number of 3DGS for each object to 1024, while Uni3D is trained on 10000 3D point clouds, which leads to an additional gap in performance. Combined with points (1) and (2), if we directly evaluate Uni3D with 1024 point clouds on Objaverse-LVIS, the Top 1 accuracy comes down to only 33.61. For our reorganized Objaverse-Lvis, it increases to 38.17.

---

> > ### Comment · Reviewer_SAh2 · 2024-11-22
> >
> > I have a small suggestion regarding the main tables (Table 1 and Table 2). Including the raw input (e.g., MV image or point cloud) available in each dataset, similar to what is shown in Figure 2, could enhance the clarity of the comparisons. This addition would help readers better understand the advantages of each method.
> >
> > Overall, I’ve reviewed the authors’ responses and found that they have addressed my concerns. Recommending accepting this paper.

---

> > > ### Author Response · Authors · 2024-11-22
> > >
> > > Thank you for your valuable feedback and suggestions. We greatly appreciate your mention of the OpenLRM project, which we will explore further to potentially enhance the broader applicability of UniGS. We will also refine the Table 1 and 2 to make the comparisons more intuitive for readers.
> > >
> > > Once again, thank you for recognizing the improvements we’ve made and for recommending our work. Your thoughtful comments have been instrumental in refining our paper, and we hope UniGS will continue to demonstrate its value in the field.

---

### Official Review · Reviewer_SvjU · 2024-11-03

**Soundness:** 2
**Presentation:** 3
**Contribution:** 2
**Rating:** 6
**Confidence:** 4

**Summary:**

This paper introduces a system for computing a joint embedding of images, text and 3D models (similar to an extension to CLIP). The paper proposes to use gaussian splats as the 3D representation instead of using point clouds to represent 3D shapes.

**Strengths:**

Building joint representations across different modalities is an interesting direction of work. Including 3D shapes can have lots of applications (which would be good to discuss in the paper).

**Weaknesses:**

I miss clear experiments that help understand the benefit of using 3D gaussian splats as the representation of 3D shapes instead of using point clouds. For instance, what numbers do you get in table 3 if you use the architecture used in UniGS but replacing the GS with point clouds?

Can you provide some scenarios in which it is useful to have the representation you propose?

The paper has no visual results. There are only tables with numbers. But it is hard to get an intuition of what the numbers mean and what the results look like just by reading those tables. It would be helpful to show some visual illustrations. You could show examples if 3D retrieval when the query is a 2D image.

**Questions:**

1. Caption in figure 2 could provide some details to help interpret the system sketch. Figure 1, figure 2 and figure 3 seem redundant. It would be clearer if they were integrated into a more comprehensive system description.

2. Section 3.1 is not very helpful, presents standard material and it is not particularly clear. Section 3.2, which introduce the architecture proposed in the paper, is short.

3. Some typos:
Line 32: “to achieve Language-Image-3D pertaining,” -> pretraining?
Line 239: “for understanding the relationships between global position and feature.” What feature?
Line 253: “We donate the process”, -> denote.

4. Section 3.4 is not very clear.  It will be helpful if figure 3 had the same notation than the one used in section 3.4. For instance, where is f_{fun}, f_{adv}? Then, in line 257 the notation changes again introducing f’_{fun} and f’_{adv}.  Why do you add the single quote ‘?

5. It would be helpful to add figures with some visual examples and comparisons across different methods. How helpful is to use GS versus point clouds?

6. The experimental results section has a lot of tables, but they are not clearly described. For instance, section 4.3 only briefly describes table 5, but it is not clear what the table shows. If this is not central to the paper, this table and section could be moved to the supplementary material section and the extra space can be used to better describe the other experiment and add a figure with visual results.

---

> ### Author Response · Authors · 2024-11-22
>
> Thank you for your careful and insightful comments. We are glad to hear that you appreciate the novelty of our proposed UniGS for multi-modal pretraining. In the following, we will address your concerns in detail:
>
> >
> >* __W1(Benefit of 3DGS over point clouds)__ : "what numbers do you get in table 3 if you use the architecture used in UniGS but replacing the GS with point clouds?"
> >
>
> | Method             | 3D Rep.       | Avg.   | Bed    | Bsf.   | Chair  | Desk   | Sofa   | Table  | Toilet  | Btub.  | Dresser | NSd.   |
> |--------------------|---------------|--------|--------|--------|--------|--------|--------|--------|---------|--------|---------|--------|
> | CLIP<sup>2</sup>   | 3DGS          | 28.50  | 1.470  | 4.000  | 40.03  | 1.640  | 15.20  | 56.72  | 4.620   | 0.000  | 26.25   | 30.51  |
> | Uni3D*             | point clouds  | 61.72  | 63.60  | **59.67** | 84.33  | **47.43** | 79.36  | **78.97** | 63.59   | 74.67  | 12.92   | 18.93  |
> | Uni3D              | 3DGS          | 54.51  | 58.09  | 19.00  | 80.38  | 17.05  | 62.40  | 47.68  | 56.92   | 48.00  | 7.500   | 11.02  |
> | Uni3D*             | 3DGS          | 56.67  | 74.63  | 28.00  | 83.89  | 28.36  | 50.88  | 54.31  | 7.690   | 20.00  | 27.50   | 19.49  |
> | UniGS              | point clouds  | 64.01  | 78.31  | 16.00  | 77.28  | 14.59  | **79.52** | 71.31  | **96.92** | 88.00  | 11.25   | 12.71  |
> | **UniGS(Ours)**    | **3DGS**      | **69.64** | **81.62** | 32.00  | **87.46** | 17.38  | 79.36  | 68.74  | 93.85   | **96.00** | **35.00** | **36.44** |
>
> **Table 1: Recognition on SUN RGBD.**  **Avg.:** the mean average Top1 accuracy across all categories. * denotes training from scratch.
>
> Thank you for your suggestion. We have conducted additional experiments in Table 1. As shown in Table 1, UniGS shows superior performance with 3DGS representation and proposed Gaussian-Aware Guidance over common point clouds, while Uni3D deteriorates due to lack of 3DGS encoder. Moreover, UniGS with point clouds still outperforms Uni3D with point clouds, demonstrating the effectiveness of UniGS in modeling explicit features of color, opacity, scale, and rotation.
>
>
> >
> >* __W2,W3,Q5(Visual comparisons)__ : "It would be helpful to show some visual illustrations. You could show examples if 3D retrieval when the query is a 2D image"
> >
>
> Thank you for the suggestion. We visualize several examples of 3D retrieval with 2D image query.
>
> As shown in Figure 8 in Sec. J of the appendix, Uni3D may mistakenly retrieve another similar object due to the similarity in point cloud structure. In contrast, UniGS demonstrates a superior 3D understanding of object color, shape, and texture with 3DGS representation and proposed Gaussian-Aware Guidance, resulting in better Image-to-3D retrieval.
>
>
> >
> >* __Q1(Figure 2 refinement)__ : "Caption in figure 2 could provide some details to help interpret the system sketch"
> >
> Thank you for the suggestion. We have refined Figure 2 in the main paper with the description of the information flow.
>
> >
> >* __Q2,Q3,Q4(Writing)__
> >
> Thank you for your careful comment. We highlight the pertaining with Language-Image-3D at line 32 to demonstrate the pertaining strategy of UniGS. The global position and features at denotes the global understanding through the position and color features of point clouds, respectively. We will clarify these typos to make it more clear.
>
> >
> >* __Q4(Figure 3 refinement)__ : "It will be helpful if figure 3 had the same notation than the one used in section 3.4"
> >
> Thank you for your careful comment. As illustrated in equations 7 and 8, $f_{fun}$ and $f_{adv}$ denote the input of the fundamental encoder and advanced encoder, respectively. We add a single quote for $f_{fun}'$ and $f_{adv}'$ to represent the features of 3D understanding from the fundamental encoder and advanced encoder. We have added the same notation to Figure 3 of the main paper for the completeness of our work.

---

> ### Author Response · Authors · 2024-11-29
>
> Thank you once again for taking the time to review our paper and for providing valuable comments to enhance its quality. We thank you for your constructive feedback, which has significantly contributed to the improvement of our work. Specifically, your comments guided us to enhance the visualizations throughout the paper, add more informative content to Figure 2, and include corresponding symbols in Figure 3 to better align with the equations. Additionally, your careful review helped us identify and correct typos in the manuscript, ensuring greater clarity and precision. Finally, your suggestion to conduct further experiments on SUN RGBD dataset has enabled us to provide a more comprehensive evaluation. Your thoughtful contributions have been invaluable in refining and strengthening our work.
>
> We sincerely hope that the additional experiments and our response have addressed your concerns. If you have any further questions or suggestions, we would be glad to offer further clarifications. Thank you for your time and consideration!

---

### Official Review · Reviewer_fg23 · 2024-11-03

**Soundness:** 3
**Presentation:** 3
**Contribution:** 3
**Rating:** 6
**Confidence:** 4

**Summary:**

UniGS introduces 3D Gaussian Splatting (3DGS) into multi-modal pre-training to address the limitations of point clouds in capturing the full complexity of 3D scenes. By representing the 3D world as collections of colored, opaque Gaussians, UniGS bridges the gap between discrete 3D points and dense 2D images. Starting from a pre-trained vision-language model, UniGS aligns 3DGS with text and image representations, creating a unified multi-modal representation. A Gaussian-Aware Guidance module further enhances fine-grained 3D feature learning and cross-modal alignment. Tested on Objaverse, ABO, MVImgNet, and SUN RGBD datasets, UniGS shows substantial improvements over the previous state-of-the-art Uni3D.

**Strengths:**

1. The improvement gain is significant compared with previous methods, which shows the effectiveness of using 3DGS as the unified 3D representation.

2. The paper introduces an innovative Gaussian-Aware Guidance module that utilizes priors from pre-trained point cloud encoders as an initialization to enhance the learning of 3DGS features. This design is effective since it doesn't require training from scratch but can make use of existing models from a different 3D representation.

3. The intuition to use 3DGS as the unified 3D representation is novel and reasonable. To the best of my knowledge, it seems to be the first paper to use 3DGS as the unified 3D representation to bridge multimodal.

**Weaknesses:**

1. Figure 2 could provide an overall description of the information flow (like how this pipeline works in general) in the caption. Also, the figure could be improved by adding some diagrams to represent downstream tasks instead of using text only.

2. I think one significant weakness of using 3DGS as a unified 3D representation is that, usually raw data doesn't use this representation, like point cloud from a Lidar sensor. In this way, this method needs to optimize or process a 3DGS using those raw data (let me know if I understand incorrectly), then leverage this unified 3D representation to conduct downstream tasks. It could be computationally expensive. It seems that this paper does not provide the computation cost for inference, I would appreciate it if authors could include this and discuss this potential weakness.

**Questions:**

1. L56-L60, the paper discusses the weakness of using point clouds as a 3D presentation due to the discrete representation. How about other potential 3D representations? Like depth maps or NeRF, which do not suffer from the discrete representation. I think some discussion among all potential 3D representations will help to justify the choice of using 3DGS as the unified 3D representation.

---

> ### Author Response · Authors · 2024-11-21
>
> Thank you for your thoughtful comments and positive recognition of our solid motivation and intuitive and nicely formulated pipeline. We are also glad that you appreciate the effectiveness of UniGS and the novelty of our Gaussian-Aware Guidance. We will address your concerns point by point:
>
> >
> >* __W1(Figure refinement)__ : "Figure 2 could provide an overall description of the information flow ... in the caption"
> >
>
> Thank you for the suggestion. We have now refined Figure 2 in the main paper with a description of the information flow.
>
> >
> >* __Q1(Comparisons to Depth- and NeRF-based approaches)__ : "the paper discusses the weakness of using point clouds as a 3D presentation due to the discrete representation. How about ... depth maps or NeRF?"
> >
>
> | Methods         | 3D Representation   | Avg.(%) ↑      |
> |------------------|---------------------|----------------|
> | **ShapeNetRender** |                     |                |
> | CLIP (1 view)    | ---                 | 73.60          |
> | CLIP (16 view)   | ---                 | 82.40          |
> | nerf2clip        | NeRF                | 84.00          |
> | nf2vec           | NeRF                | 87.30          |
> | Uni3D            | 3DGS location       | 88.96          |
> | **UniGS (Ours)** | 3DGS                | **93.94**      |
>
> **Table 3**: Zero-shot classification on ShapeNetRender. Avg.: the mean average Top1 classification accuracy. Uni3D and UniGS are trained for 15 epochs on ShapeNetRender.
>
> | Method          | 3D Rep.      | Avg.   | Bed    | Bsf.   | Chair  | Desk   | Sofa   | Table  | Toilet | Btub.  | Dresser | NSd.   |
> |------------------|--------------|--------|--------|--------|--------|--------|--------|--------|--------|--------|---------|--------|
> | CLIP${^2}$      | 3DGS         | 28.50  | 1.470  | 4.000  | 40.03  | 1.640  | 15.20  | 56.72  | 4.620  | 0.000  | 26.25   | 30.51  |
> | Uni3D*          | Point Clouds | 61.72  | 63.60  | **59.67** | 84.33  | **47.43** | **79.36** | **78.97** | 63.59  | 74.67  | 12.92   | 18.93  |
> | Uni3D           | 3DGS         | 54.51  | 58.09  | 19.00  | 80.38  | 17.05  | 62.40  | 47.68  | 56.92  | 48.00  | 7.500   | 11.02  |
> | Uni3D*          | 3DGS         | 56.67  | 74.63  | 28.00  | 83.89  | 28.36  | 50.88  | 54.31  | 7.690  | 20.00  | 27.50   | 19.49  |
> | PointClip       | Depth        | 11.50  | 0.000  | 94.00  | 0.000  | 0.000  | 0.000  | 14.70  | 0.000  | 0.000  | 6.100   | 0.000  |
> | PointClip*      | Depth        | 38.00  | 45.30  | 100.0  | 62.50  | 48.50  | 44.40  | 4.800  | 55.20  | 16.30  | 3.300   | 0.000  |
> | Clip2Point      | Depth        | 18.60  | 10.90  | 20.60  | 64.30  | 34.40  | 13.80  | 14.10  | 26.20  | 0.000  | 1.400   | 0.00   |
> | Clip2Point*     | Depth        | 56.90  | 78.00  | 87.60  | 36.20  | 36.60  | 64.70  | 37.40  | 82.10  | 77.50  | **67.60** | 1.20   |
> | **UniGS (Ours)** | 3DGS         | **69.64** | **81.62** | 32.00  | **87.46** | 17.38  | **79.36** | 68.74  | **93.85** | **96.00** | 35.00   | **36.44** |
>
> **Table 4**: **Recognition on SUN RGBD.**  **Avg.**: The mean average Top1 accuracy across all categories. * denotes training from scratch.
>
>
> Thank you for your suggestion. We have conducted additional comparisons to NeRF-based[1,2] and Depth-based[3,4] approaches. As shown in Table 3, UniGS outperforms nerf2clip[1] and nf2vec[2] with 9.93\% and 6.64\%, demonstrating significant improvement over NeRF-based approaches on cross-modalities learning.
>
> Moreover, we supply comparisons to Depth-based approaches and reformulate the results in Table 4. As illustrated in Table 4, UniGS significantly outperforms PointCLIP[3] and Clip2Point[4] on the SUN RGBD datasets with over 31.64\% and 12.74\%, demonstrating the effectiveness of the 3DGS representation.
>
> [1] Connecting NeRFs, Images, and Text
>
> [2] Deep learning on 3D neural fields.
>
> [3] Learning transferable visual models from natural language supervision
>
> [4] Clip2point: Transfer clip to point cloud classification with image-depth pre-training

---

> ### Author Response · Authors · 2024-11-21
>
> >
> >* __W2(Computational cost)__ : "It seems that this paper does not provide the computation cost for inference"
> >
> | Methods       | FLOPs(G) ↓ | Time(ms) ↓ | Top 1 Avg.   |
> |---------------|------------|------------|--------------|
> | CLIP²         | 22.49      | 232        | 10.20        |
> | TAMM          | 22.49      | 233        | 22.70        |
> | Uni3D         | 47.85      | 113        | 30.47        |
> | UniGS(Ours)   | 98.17      | 233        | **38.57**    |
>
> **Table 1**: Comparisons of forward computational cost on Objaverse-Lvis.
>
>
> __`Runtime analysis`__: Thank you for the comment. We have added an evaluation of the FLOPs and runtime of UniGS and compared them with state-of-the-art approaches. In the results in Table 1, observe that with a slight increase in runtime, UniGS achieves significant improvement over ${CLIP}^{2}$, TAMM, and Uni3D on Objaverse-Lvis zero-shot classification.
>
> | Fundamental Encoder |            | Advanced Encoder |            | Cross-Attn | Others | FLOPs  |
> |---------------------|------------|------------------|------------|------------|--------|--------|
> | CNN layers          | ViT blocks| CNN layers       | ViT blocks |            |        |        |
> | ✓                   |          |                 |           |           |       | 36.67  |
> | ✓                   | ✓         |                 |           |           |       | 47.60  |
> | ✓                   | ✓         | ✓                |           |           |       | 84.31  |
> | ✓                   | ✓         | ✓                | ✓          |           |       | 95.24  |
> | ✓                   | ✓         | ✓                | ✓          | ✓          |       | 95.43  |
> | ✓                   | ✓         | ✓                | ✓          | ✓          | ✓      | 95.94  |
>
> **Table 2**: Ablation study on the FLOPs of UniGS modules. CNN encoder denotes the CNN layers to extract spatial information from 3D representation into features, and ViT blocks denotes the Transformer blocks understanding objects from extracted features. Cross-Attn denotes the Cross-attention layers between Fundamental and Advanced Encoder.
>
> Moreover, as shown in the Table 2 and Figure 7 in Sec. J of the appendix, we further conduct an in-depth evaluation of the FLOPs for our UniGS framework, which provides a better understanding of the increased computational cost. Specifically, __76.5\% of the total FLOPs (73.38G) is spent in the CNN layers of the 3D Encoder to extract 3D spatial features__ for understanding.
> This is indeed a limitation of our method's general application.
> Fortunately, much progress is being made in compressing models [1] and 3D representations [2], and we expect these advances to facilitate the development of 3D understanding with 3DGS representation.
>
>
> __`Time analysis of 3DGS optimization`__: Indeed, 3DGS involves additional time for optimization and the details of the required 3DGS optimization cost are discussed in Line at line 740 in the main paper. Specifically, preparing 800k 3DGS objects with 1024 3D Gaussian kernels takes a week to prepare, while 800k 3DGS objects with 10000 3D Gaussian kernels takes about 12 days. The preparation of 3DGS datasets demands significant efforts in terms of time and computational power. We will therefore make all prepared datasets publicly available to the community to support further advancements in 3DGS representation learning.
>
> Moreover, leveraging image-to-3DGS approaches for dataset preparation is another promising step. For example, GS-LRM [3] and PF-LRM [4] takes only 0.23 and 1.3 seconds from 2-4 posed sparse images to generate 3DGS, respectively. Although recent advances have been made on image-to-3DGS [3,4], they are unfortunately still not open-sourced now. Given the popularity of 3DGS, we expect these representations to only become more and more efficient to construct.
>
> __`Compatability with raw data representations`__: As illustrated in Table 5 of the main paper, 3DGS showcases potential compatibility with point clouds, such that the proposed approach can leverage both the processed 3DGS representations jointly with the raw data representations, offering promise for practical applications in real-world scenarios.
>
> [1] T3DNet: Compressing Point Cloud Models for Lightweight 3D Recognition
>
> [2] LightGaussian: Unbounded 3D Gaussian Compression with 15x Reduction and 200+ FPS
>
> [3] GS-LRM: Large Reconstruction Model for 3D Gaussian Splatting
>
> [4] PF-LRM: Pose-Free Large Reconstruction Model for Joint Pose and Shape Prediction

---

> ### Author Response · Authors · 2024-11-28
>
> Thank you once again for taking the time to review our paper and for providing valuable comments to enhance its quality. We appreciate your thoughtful comments, which have guided us in improving our work. Specifically, we have refined Figure 2 by incorporating information flow to improve clarity, added comparisons with NeRF- and Depth-based methods to enhance the completeness of the paper, and conducted additional experiments on computational cost and ablation studies to explore the potential for future applications in lightweight scenarios. Your insightful suggestions have been invaluable in helping us address these aspects comprehensively.
>
> We sincerely hope that the additional experiments and our response have addressed your concerns. If you have any further questions or suggestions, we would be glad to offer further clarifications. Thank you for your time and consideration!

---

### Official Review · Reviewer_yxJz · 2024-11-05

**Soundness:** 3
**Presentation:** 3
**Contribution:** 2
**Rating:** 6
**Confidence:** 4

**Summary:**

This paper presents a text-image-3D pre-training framework that leverages 3DGS as the 3D representation for multi-modal representation. Experiments on various datasets are conducted.

**Strengths:**

- The proposed method has good performance on various multi-modal datasets.
- The experimental results are given on two different datasets COCO and VisDrone.

**Weaknesses:**

- Uni3D has used one model to unify the 3D representations from different models, which can be used to align with image and text. What is the main advantage of proposed method using 3DGS.
- The proposed method introduces 3DGS for feature experimentation. Does it increase computational cost.
- It seems that most experiment are not inconsistent with the results in Uni3D. In this paper, the performance are relatively poor. What is the difference.
- I think that it is better to use the similar settings as Uni3D for experiments.

**Questions:**

I suggest that the authors provide more experiments and illustrations for the questions in weakness

---

> ### Author Response · Authors · 2024-11-20
>
> Thank you for your careful and insightful comments. We are glad to hear that you appreciate the effectiveness of our proposed UniGS for multi-modal pretraining. In the following, we will address your concerns in detail:
>
> >
> >* __W1(Advantage of 3DGS)__ : "What is the main advantage of proposed method using 3DGS"
> >
> Thank you for the comment. As shown in Figure 6 in Sec. J of the appendix, we highlight the difference between Uni3D and UniGS. Our UniGS is also capable of utilizing __a single model to unify 3D representations from different models__, with better performance with 3DGS representation and proposed Gaussian-Aware Guidance.
> Specifically, when using point clouds as a unified 3D representation, the main challenge is __the divergence between the 3D representation and other modalities__.
> In contrast, UniGS leverages 3DGS as the 3D representation, which effectively reconstructs the 3D target object as well as provides efficient correspondences between 3D and 2D images.
>
> Through extensive experiments across the Objaverse, ABO, MVImgNet and SUN RGBD datasets with zero-shot classification, text-driven retrieval, and open-world understanding tasks, we demonstrate the effectiveness of UniGS in learning a more general and stronger aligned multi-modal representation.
>
>
> >
> >* __W2(Computational cost)__ : "Does it increase computational cost."
> >
> | Methods       | FLOPs(G) ↓ | Time(ms) ↓ | Top 1 Avg.   |
> |---------------|------------|------------|--------------|
> | CLIP²         | 22.49      | 232        | 10.20        |
> | TAMM          | 22.49      | 233        | 22.70        |
> | Uni3D         | 47.85      | 113        | 30.47        |
> | UniGS(Ours)   | 98.17      | 233        | **38.57**    |
>
> **Table 1**: Comparisons of forward computational cost on Objaverse-Lvis.
>
>
> Thank you for the comment. We further evaluate the FLOPs and runtime of UniGS and compare them with state-of-the-art approaches in Table 1. With a slight increase in runtime, UniGS achieves significant improvement over ${CLIP}^{2}$, TAMM, and Uni3D on Objaverse-Lvis zero-shot classification.
>
> | Fundamental Encoder |            | Advanced Encoder |            | Cross-Attn | Others | FLOPs  |
> |---------------------|------------|------------------|------------|------------|--------|--------|
> | CNN layers          | ViT blocks| CNN layers       | ViT blocks |            |        |        |
> | ✓                   |          |                 |           |           |       | 36.67  |
> | ✓                   | ✓         |                 |           |           |       | 47.60  |
> | ✓                   | ✓         | ✓                |           |           |       | 84.31  |
> | ✓                   | ✓         | ✓                | ✓          |           |       | 95.24  |
> | ✓                   | ✓         | ✓                | ✓          | ✓          |       | 95.43  |
> | ✓                   | ✓         | ✓                | ✓          | ✓          | ✓      | 95.94  |
>
> **Table 2**: Ablation study on the FLOPs of UniGS modules. CNN encoder denotes the CNN layers to extract spatial information from 3D representation into features, and Trans. denotes the Transformer blocks understanding objects from extracted features. Cross-Attn denotes the Cross-attention layers between Fundamental and Advanced Encoder.
>
> Moreover, to better understand the computational cost of 3D-based approaches, we present an additional ablation study of UniGS modules on FLOPs in Table 2 and Figure 7 in Sec. J of the appendix. Specifically, this helps us understand the difference as __76.5\% of the total FLOPs (73.38G) is due to the CNN layers of the 3D Encoder to extract 3D spatial features__.
> This is indeed a limitation of our method's general application.
> Fortunately, much progress is being made in compressing models [1] and 3D representations [2], and we expect these advances to facilitate the development of 3D understanding with 3DGS representation.
>
> [1] T3DNet: Compressing Point Cloud Models for Lightweight 3D Recognition
>
> [2] LightGaussian: Unbounded 3D Gaussian Compression with 15x Reduction and 200+ FPS

---

> ### Author Response · Authors · 2024-11-20
>
> >
> >* __W3,W4(Comparison with Uni3D under the same setting)__ : "it is better to use the similar settings as Uni3D for experiments."
> >
>
> | Methods | Backbone            | Top1   | Top3   | Top5   | Representation  |
> |---------|---------------------|--------|--------|--------|-----------------|
> | **10000 3D points**           |                     |        |        |        |                 |
> | Uni3D   | EVA02-S-patch14     | 50.34  | 72.70  | 79.81  | point clouds    |
> | UniGS   | EVA02-S-patch14     | **53.16** | **75.59** | **82.14** | 3DGS            |
>
> **Table 3**: Comparison results with 10000 points dataset on Objaverse-Lvis zero-shot classification.
>
> Thank you for the suggestion. UniGS outperforms Uni3D under a fair setting of representing objects with 1024 3D points and training with CLIP ViT-B-16. For comprehensive comparisons, we conduct further comparisons with the same setting of Uni3D and present the experiment results in Table 3. __Under the same setting of Uni3D__ with a higher number of 3D gaussian kernels representing objects, __UniGS still outperform Uni3D on Objaverse-Lvis zero-shot classification__.

---

> ### Author Response · Authors · 2024-11-25
>
> Thank you once again for taking the time to review our paper and for providing valuable comments to enhance its quality. As the deadline for reviewer-author discussions draws near, we look forward to your feedback on our response. If you have any additional comments, we would be glad to offer further clarifications. Thank you!

---

> > ### Comment · Reviewer_yxJz · 2024-11-27
> >
> > The rebuttal has solved most of my concerns. I think it is necessary to add the related experiments in the paper, such as Table 1 and Table 3 here.

---

> ### Author Response · Authors · 2024-11-27
>
> Thank you for your valuable feedback and suggestions. We appreciate your instrumental comments in refining our paper, and we have added the additional experiments in the appendix for the completeness of our work. We are glad that most of your concerns have been resolved. If you have any additional comments, we would be glad to offer further clarifications. Thank you!

---

> > ### Comment · Reviewer_yxJz · 2024-11-27
> >
> > I donot see the experiments, such as Table 1 and Table 3, in the paper.

---

> ### Author Response · Authors · 2024-11-27
>
> Sorry for this unclarity. We have provided Table 1 and Table 3 in Section J of the appendix (please refer to Tables 13, 14, and 15), with the differences highlighted in red. We believe the additional experiments can be found in the latest updated PDF and we hope this can address your concern.

---

> > ### Comment · Reviewer_yxJz · 2024-11-27
> >
> > Thanks. I get it and change my rating.

---

### Official Review · Reviewer_agiJ · 2024-11-08

**Soundness:** 3
**Presentation:** 3
**Contribution:** 3
**Rating:** 6
**Confidence:** 3

**Summary:**

This paper presents a novel multi-modal pre-training method, UniGS, designed to achieve more general and efficient joint representations of text, images, and point clouds. The innovative introduction of 3D Gaussian Splatting (3DGS) and the proposed Gaussian-Aware Guidance module achieve leading results in different 3D tasks. The experiments are solid and can confirm the effectiveness of the proposed method.

**Strengths:**

The proposed approach can achieve state-of-the-art performance on various challenging datasets, which demonstrates the effectiveness in learning strong cross-model representations.

**Weaknesses:**

1）No detailed explanation was given on how negative samples were selected. Do different tasks need to adjust the negative sample strategy?
2）There are significant differences in the structural and spatial characterization of 3DGS and traditional point cloud data. Does it affect the final performance？

**Questions:**

See weakness.

---

> ### Author Response · Authors · 2024-11-21
>
> Thank you for your detailed and thoughtful feedback on our paper. We are pleased that you recognize the strengths of UniGS, including the state-of-the-art performance and the effectiveness of 3DGS cross-model representations in cross-model learning. We will address your concerns point by point:
>
> >
> >* __W1(Details of negative sample strategy)__ : "Do different tasks need to adjust the negative sample strategy?"
> >
>
> | MoCo   | Momentum Steps | Top 1   | Epoch | Text-image Model | Embedding Dim |
> |--------|----------------|---------|-------|------------------|---------------|
> | ✗      | ---            | 47.10   | 15    | ViT-B-16         | 512           |
> | ✓      | 1              | 48.06   | 20    |                  |               |
> | ✓      | 3              | 46.24   | 20    |                  |               |
> | ✓      | 5              | **50.37** | 25   |                  |               |
> |--------|----------------|---------|-------|------------------|---------------|
> | ✗      | ---            | 53.07   | 15    | Laion-H          | 768           |
> | ✓      | 1              | 52.51   | 15    |                  |               |
> | ✓      | 3              | **54.24** | 20   |                  |               |
> | ✓      | 5              | 54.07   | 25    |                  |               |
>
> **Table 1**: **Ablation study on the schedule of negative sampling.** MoCo denotes *Momentum Contrast for Unsupervised Visual Representation Learning*.
>
>
>
> Thank you for the comment. During training, batches of data are randomly sampled from the dataset, and will be gathered between GPUs when computing loss. For a single object, its positive image sample is its corresponding image, while its negative samples are derived from images of other objects. Similarly, for text data, the positive sample is the object's own corresponding text, whereas the negative samples are taken from other texts that differ from its corresponding text. __This random negative sampling strategy works well across different downstream tasks.__
>
> Moreover, our framework is open to improvements in the negative sampling strategy. As shown in Table 1, we apply MoCo[1] as a better strategy of negative sampling for UniGS. __Given more training iterations, UniGS can achieve better Top 1 accuracy with MoCo for different text-image models.__
>
> [1] Momentum Contrast for Unsupervised Visual Representation Learning
>
> >
> >* __W2(Impact of structural differences)__ : "There are significant differences in the structural and spatial characterization of 3DGS and traditional point cloud data. Does it affect the final performance?"
> >
> | Methods      | Backbone             | Top 1   | Top 3   | Top 5   | Representation | Augment w. point clouds |
> |--------------|----------------------|---------|---------|---------|-----------------|--------------------------|
> | **10000 3D points**                                                                             |
> | Uni3D        | EVA02-S-patch14      | 50.34   | 72.70   | 79.81   | point clouds    | ---                      |
> | UniGS        | EVA02-S-patch14      | 52.44   | 75.37   | **82.71** | 3DGS           | ✓                        |
> | UniGS†       | EVA02-S-patch14      | **53.16** | **75.59** | 82.14 | 3DGS           | ✗                        |
>
> **Table 2**: **Comparison results with 10000 points dataset on Objaverse-Lvis zero-shot classification.**  † denotes fine-tuning on 3DGS datasets.
>
> Thank you for your suggestion. We conducted additional experiments to fine-tune UniGS using a pure 3DGS dataset. As shown in Table 2, __UniGS outperforms Uni3D when augmented with point clouds under the same settings as Uni3D__. It also achieves higher Top-1 and Top-3 accuracy after fine-tuning on the pure 3DGS dataset. Experimental results in Table 2 and the main paper further demonstrate that Uni3D is not fully compatible with both point clouds and 3DGS. However, with our proposed Gaussian-Aware Guidance, __UniGS exhibits the ability to effectively understand objects in both point clouds and 3DGS__, achieving superior results after fine-tuning.

---

> ### Author Response · Authors · 2024-11-27
>
> Thank you once again for taking the time to review our paper and for providing valuable comments to enhance its quality. We appreciate your insightful comments on improving UniGS by employing an enhanced negative samples strategy and fine-tuning on a pure 3DGS dataset. We are grateful for your thoughtful feedback to explore the best performance of UniGS.
>
> We sincerely hope that the additional experiments and our response have addressed your concerns. If you have any further questions or suggestions, we would be glad to offer further clarifications. Thank you for your time and consideration!

---

### Author Response · Authors · 2024-11-22
**General Response – Thanks to All Reviewers for Constructive and Insightful Feedback**

We would like to thank all reviewers for their positive affirmations on the novelty and potential impact of this paper, which leverages 3DGS as the 3D representation for learning a more general and stronger multi-modal representation and proposes a novel Gaussian-Aware Guidance module to leverage priors from pre-trained point cloud encoders for better 3D understanding.

Reviewer agiJ found the innovative introduction of 3D Gaussian Splatting and the effectiveness of the proposed Gaussian-Aware Guidance module in different 3D tasks, emphasizing our work is solid and credible. Reviewer yxjz recognized the effectiveness of our proposed framework of text-image-3D pre-training with 3DGS representation. Reviewer fg23 noted that our framework proposes a unique Gaussian-Aware Guidance module for improving 3D comprehension and achieving state-of-the-art performance, emphasizing the intuition and novelty of the article. Reviewer SvjU appreciated the introduction of our approach for aligning 3DGS representation, which is potentially significant for future real-world representation learning. Reviewer SAh2 highlighted that our paper introduces 3D Gaussian Primitives for alignment with Vision-Language Models via Pretraining, achieving state-of-the-art performance with the novel Gaussian-Aware Guidance module.

We believe that we have been able to thoroughly address the Reviewers’ comments by clarifying certain sections of the paper and incorporating additional experiments. Details on these changes can be found in the response to individual comments by the Reviewers.

Specifically, to address the concerns raised by Reviewer yxJz, we have

1. Included more comparisons of computational cost on FLOPs and runtime.
2. Included more ablation study results on the computational cost of UniGS.
3. Included more evaluation results to understand the same setting with Uni3D.

Additionally, we will open-source all the code and the prepared large-scale datasets to contribute to further research in this area.

---

### Meta-Review · Area_Chair_pVeQ · 2024-12-22

**Metareview:**

The manuscript received positive ratings (6, 6, 6, 6, and 6). Reviewers appreciated the intuition to use 3DGS as the unified 3D representation, the design of the Gaussian-Aware Guidance module, and the performance improvements obtained by the proposed approach on different datasets. Reviewers also raised several concerns in the initial review including, the computational cost, more explanations with respect to results reported in Uni3D, additional ablation regarding using the architecture employed in UniGS but replacing the GS with point clouds, additional comparisons with NeRF-based approaches, and more visual results. Authors provided a rebuttal to address the concerns of the reviewers including, additional details and results regarding negative sample strategy, computational cost comparison, comparisons with NeRF- and Depth-based approaches, additional ablation analysis, and clarification on visual results. Reviewers expressed that most of their concerns are addressed in the rebuttal and remained positive about the manuscript. Given the reviewers comments, rebuttal and discussions, the recommendation is accept. Authors are strongly encouraged to take into consideration reviewers feedback when preparing the revised manuscript.

**Additional Comments On Reviewer Discussion:**

Reviewers raised several concerns in the initial review including, the computational cost, more explanations with respect to results reported in Uni3D, additional ablation regarding using the architecture employed in UniGS but replacing the GS with point clouds, additional comparisons with NeRF-based approaches, and more visual results. Authors provided a rebuttal to address the concerns of the reviewers including, additional details and results regarding negative sample strategy, computational cost comparison, comparisons with NeRF- and Depth-based approaches, additional ablation analysis, and clarification on visual results. Reviewers expressed that most of their concerns are addressed in the rebuttal and remained positive about the manuscript.

---

### Decision · Program_Chairs · 2025-01-22

Accept (Poster)